# Cross-scale interaction of host tree size and climatic water deficit governs bark beetle-induced tree mortality

Michael J. Koontz [1,2,3✉], Andrew M. Latimer [1,2], Leif A. Mortenson[4], Christopher J. Fettig[5] & Malcolm P. North[1,2,6]

The recent Californian hot drought (2012–2016) precipitated unprecedented ponderosa pine (*Pinus ponderosa*) mortality, largely attributable to the western pine beetle (*Dendroctonus brevicomis*; WPB). Broad-scale climate conditions can directly shape tree mortality patterns, but mortality rates respond non-linearly to climate when local-scale forest characteristics influence the behavior of tree-killing bark beetles (e.g., WPB). To test for these cross-scale interactions, we conduct aerial drone surveys at 32 sites along a gradient of climatic water deficit (CWD) spanning 350 km of latitude and 1000 m of elevation in WPB-impacted Sierra Nevada forests. We map, measure, and classify over 450,000 trees within 9 km$^2$, validating measurements with coincident field plots. We find greater size, proportion, and density of ponderosa pine (the WPB host) increase host mortality rates, as does greater CWD. Critically, we find a CWD/host size interaction such that larger trees amplify host mortality rates in hot/dry sites. Management strategies for climate change adaptation should consider how bark beetle disturbances can depend on cross-scale interactions, which challenge our ability to predict and understand patterns of tree mortality.

[1] Graduate Group in Ecology, University of California, Davis, CA, USA. [2] Department of Plant Sciences, University of California, Davis, CA, USA. [3] Earth Lab, University of Colorado-Boulder, Boulder, CO, USA. [4] USDA Forest Service, Pacific Southwest Research Station, Placerville, CA, USA. [5] USDA Forest Service, Pacific Southwest Research Station, Davis, CA, USA. [6] USDA Forest Service, Pacific Southwest Research Station, Mammoth Lakes, CA, USA. ✉email: michael.koontz@colorado.edu

Bark beetles dealt the final blow to many of the nearly 150 million trees killed in the California hot drought of 2012–2016 and its aftermath[1]. A harbinger of climate change effects to come, record high temperatures exacerbated the drought[2,3], which increased water stress in trees[4,5], making them more susceptible to colonization by bark beetles[6,7]. Further, a century of fire suppression has enabled forests to grow into dense stands, which can also make them more vulnerable to bark beetles[6,8,9]. This combination of environmental conditions and forest structural characteristics led to tree mortality events of unprecedented size across the state[10,11].

Tree mortality exhibited a strong latitudinal and elevational gradient[4,11] that can only be partially explained by coarse-scale measures of environmental conditions (i.e., historic climatic water deficit; CWD) and current forest structure (i.e., current regional basal area)[11]. A progressive loss of canopy water content offers additional insight into tree stress and mortality risk, but cannot ultimately resolve which trees are actually killed by bark beetles or elucidate factors driving bark beetle population dynamics and spread[5]. Bark beetles respond to local forest characteristics in positive feedbacks that non-linearly alter tree mortality dynamics against a background of environmental conditions that stress trees[12,13]. Thus, explicit consideration of local forest structure and composition[14,15], as well as its cross-scale interaction with regional climate conditions[16], can refine our understanding of tree mortality patterns from California's recent hot drought. The challenge of simultaneously measuring the effects of both local-scale forest features (such as structure and composition) and broad-scale environmental conditions (e.g., CWD) on forest insect disturbance leaves their interaction effect relatively underexplored[14–17].

The ponderosa pine/mixed-conifer forests in California's Sierra Nevada region are characterized by regular bark beetle disturbances, primarily by the influence of western pine beetle (*Dendroctonus brevicomis*; WPB) on its host ponderosa pine (*Pinus ponderosa*)[18]. WPB is a primary bark beetle—its reproductive success is contingent upon host tree mortality, which itself requires enough beetles to mass attack the host tree and overwhelm its defenses[19]. This Allee effect creates a strong coupling between beetle selection behavior of host trees and host tree susceptibility to colonization[19–21]. A key defense mechanism of conifers to bark beetle attack is to flood beetle boreholes with resin, which physically expels colonizing beetles, can be toxic to the colonizers and their fungi, and may interrupt beetle communication[22,23]. Under normal conditions, weakened trees with compromised defenses are the most susceptible to colonization and will be the main targets of primary bark beetles like WPB[13,23,24]. Under severe water stress, however, many trees no longer have the resources available to mount a defense[7,13]. Drought[12,25–27], especially when paired with high temperatures[24,28–30], can trigger increased bark beetle-induced tree mortality as average tree vigor declines. As the local population density of beetles increases due to successful reproduction within spatially aggregated susceptible trees, mass attacks grow in size and become capable of overwhelming formidable tree defenses. Even large healthy trees may be susceptible to colonization and mortality when beetle population density is high[13,23,24]. Thus, water stress and beetle population density interact to influence whether individual trees are susceptible to bark beetles. When extreme or prolonged drought increases host tree vulnerability, bark beetle population growth rates increase, then become self-amplifying as greater beetle densities make additional host trees prone to successful mass attack[12,13,15,24].

WPB activity is strongly influenced by forest structure—the spatial arrangement and size distribution of trees—and tree species composition. Taking forest structure alone, high-density forests are more prone to bark beetle-induced tree mortality compared to thinned forests[6,9], which may arise as greater competition for water resources amongst crowded trees lowers average tree resistance[31], or because smaller gaps between trees protect pheromone plumes from dissipation by the wind and thus enhance intraspecific beetle communication[32]. Tree size is another aspect of forest structure that affects bark beetle host selection behavior with smaller trees tending to have a lower capacity for resisting attack, but larger trees being more desirable targets on account of their thicker phloem providing greater nutritional content[13,33–35]. Throughout an outbreak, some bark beetle species will collectively "switch" the preferred size of the tree to attack in order to navigate this trade-off between host susceptibility and host quality[13,21,36–39]. Taking forest composition alone, WPB activity in the Sierra Nevada mountain range of California is necessarily tied to the regional distribution of its exclusive host, ponderosa pine[18]. Colonization by primary bark beetles can also depend on the local relative frequencies of tree species in forest stands, reflecting the more general pattern that specialist insect herbivory tends to be lower in taxonomically diverse forests compared to monocultures[40,41].

The interaction between forest structure and composition at both stand- and tree-scales also drives WPB activity. For instance, dense forest stands with high host availability may experience greater beetle-induced tree mortality because dispersal distances between potential host trees are shorter, which reduces predation of adults searching for hosts and facilitates higher rates of colonization[33,42,43]. High host availability can also reduce the chance of individual beetles wasting their limited resources flying to and landing on a non-host tree[44,45]. At a finer scale, a host tree's defensive capacity can depend on its canopy position, with reduced biochemical defenses in suppressed, crowded trees[46]. Coarse-scale measures of forest structure and composition can therefore only partially explain mechanisms affecting bark beetle disturbance. Finer-grain information is also needed that explicitly recognizes tree species, size, and local density, which better captures the ecological processes underlying insect-induced tree mortality[28,36,38,39].

The vast spatial extent of WPB-induced tree mortality in the 2012–2016 California hot drought challenges our ability to simultaneously consider how broad-scale environmental conditions may interact with local forest structure and composition to affect the dynamic between bark beetle selection and colonization of host trees, and host tree susceptibility to attack[15,47]. Measuring local forest structure generally requires expensive instrumentation[4,48] or labor-intensive field surveys[14,15,49], which constrains survey extent and frequency. Small, unhumanned aerial systems (sUAS) enable relatively fast and cheap remote imaging over hundreds of hectares of forest, which can be used to measure complex forest structure and composition at the individual tree scale with Structure from Motion (SfM) photogrammetry[50,51]. The ultra-high, centimeter-scale resolution of sUAS-derived measurements, as well as the ability to incorporate vegetation reflectance, can help overcome challenges in species classification and dead tree detection inherent in other remote sensing methods, such as airborne LiDAR[52]. Distributing such surveys across an environmental gradient can overcome the data acquisition challenge inherent in investigating phenomena with both a strong local- and a strong broad-scale component.

We used sUAS-derived remote sensing images over a network of 32 sites in Sierra Nevada ponderosa pine/mixed-conifer forests spanning 1000 m of elevation and 350 km of latitude[14] covering a total of 9 km$^2$, to investigate how broad-scale environmental conditions interacted with local forest structure and composition to shape patterns of tree mortality during the cumulative tree mortality event of 2012 to 2018. We asked:

**Table 1 Correlation and differences between the best-performing tree detection algorithm (lmfx with dist2d = 1 and ws = 2.5) and the ground data.**

| Forest structure metric | Ground mean | Correlation with ground | RMSE | Median error |
|---|---|---|---|---|
| Total tree count | 19 | 0.67* | 8.68* | 2 |
| Count of trees >15 m | 9.9 | 0.43 | 7.38 | 0 |
| Distance to 1st neighbor (m) | 2.8 | 0.55* | 1.16* | 0.26 |
| Distance to 2nd neighbor (m) | 4.3 | 0.61* | 1.70* | 0.12 |
| Height (m); 25th percentile | 12 | 0.16 | 8.46 | −1.2 |
| Height (m); mean | 18 | 0.29 | 7.81* | −2.3 |
| Height (m); 75th percentile | 25 | 0.35 | 10.33* | −4 |

An asterisk next to the correlation or RMSE indicates that this value was within 5% of the value of the best-performing algorithm/parameter set. Ground mean represents the mean value of the forest metric across the 110 field plots that were visible from the sUAS-derived imagery. The median error is calculated as the median of the differences between the air and ground values for the 110 visible plots. Thus, a positive number indicates an overestimate by the sUAS workflow and a negative number indicates an underestimate.

1. How does the proportion of the ponderosa pine host trees in a local area and average host tree size affect WPB-induced tree mortality?
2. How does the density of all trees (hereafter "overall density") affect WPB-induced tree mortality?
3. How does the total basal area of all trees (hereafter "overall basal area") affect WPB-induced tree mortality?
4. How does environmentally driven tree moisture stress affect WPB-induced tree mortality?
5. How do the effects of forest structure, forest composition, and environmental condition interact to influence WPB-induced tree mortality?

Here, we show that a greater local proportion of host trees (ponderosa pine) strongly increases the probability of host mortality, with greater host density amplifying this effect. We also show that greater site-level CWD increases host mortality rates. Further, we show that larger host trees increase the probability of host mortality in accordance with the well-known life history of WPB. Critically, we find a strong interaction between host size and CWD such that host mortality rates are especially high in hot/dry sites where the local average host tree size is large. Our results demonstrate a cross-scale interaction in the response of WPB to local forest structure and composition across an environmental gradient, which helps reconcile differences between observed ecosystem-wide tree mortality patterns and predictions from models based on coarser-scale forest structure.

## Results

**Tree detection algorithm performance.** We found that the experimental lmfx algorithm[53] with parameter values of dist2d = 1 and ws = 2.5 performed the best across seven measures of forest structure as measured by Pearson's correlation with ground data (Table 1).

**Classification accuracy for live/dead and host/non-host.** The accuracy of live/dead classification on a withheld testing data set was 96.4%. The accuracy of species classification on a withheld testing data set was 64.1%. The accuracy of WPB host/non-WPB-host (i.e., ponderosa pine versus other tree species) on a withheld testing data set was 71.8%.

**Site summary based on best tree detection algorithm and classification.** Across all study sites, we detected, segmented, and classified 452,413 trees in 23,187, 20 × 20 m pixels (with the area of each pixel being approximately equivalent to that of a field plot). Of these trees, we classified 118,879 as dead (26.3% mortality). Estimated site-level tree mortality ranged from 6.8% to 53.6%. See Supplementary Table 1 for site summaries and comparisons to site-level mortality measured from field data.

**Effect of local structure and regional climate on tree mortality attributed to WPB.** Site-level CWD exerted a positive main effect on the probability of ponderosa mortality (effect size: 0.85; 95% CI: [0.70, 0.99]; Fig. 1). We found a positive main effect of the proportion of host trees per cell (effect size: 0.68; 95% CI: [0.62, 0.74]), with a greater proportion of host trees (i.e., ponderosa pine) in a cell increasing the probability of ponderosa pine mortality. We detected no effect of overall tree density or overall basal area (i.e., including both ponderosa pine and non-host species; tree density effect size: −0.01; 95% CI: [−0.11, 0.08]; basal area effect size: −0.13; 95% CI: [−0.29, 0.03]).

We found a positive two-way interaction between the overall tree density per cell and the proportion of trees that were hosts, which is equivalent to a positive effect of the density of host trees (effect size: 0.06; 95% CI: [0.01, 0.12]; Fig. 1).

We found a positive main effect of the mean height of ponderosa pine on the probability of ponderosa mortality (effect size: 0.25; 95% CI: [0.14, 0.35]). Coupled with the strong correlation between the proportion of dead host trees and basal area killed (Supplementary Fig. 1 and Supplementary Note 1), these results suggest that WPB-attacked larger trees, on average. Further, there was a strong positive interaction between CWD and ponderosa pine mean height, such that larger trees were especially likely to increase the local probability of ponderosa mortality in hotter, drier sites (effect size: 0.54; 95% CI: [0.37, 0.70]; Fig. 2).

We found no effect of the site-level CWD interactions with the proportion of host trees (effect size: −0.08; 95% CI: [−0.18, 0.03]) or of the interaction between CWD and total basal area (effect size: −0.04; 95% CI: [−0.23, 0.15]; Fig. 1).

We found a negative effect of the CWD interaction with overall tree density (effect size: −0.19; 95% CI: [−0.31, −0.07]) as well as of the interaction between the mean height of host trees and the overall basal area (effect size: −0.08; 95% CI: [−0.13, −0.03]; Fig. 1).

While we found no interaction between the proportion of host trees and mean host tree height, we did find a 3-way interaction between these variables with CWD (effect size: 0.14; 95% CI: [0.04, 0.24]; Fig. 1).

## Discussion

This study uses drone-derived imagery to refine our understanding of the patterns of tree mortality following the 2012 to 2016 California hot drought and its aftermath. By simultaneously measuring the effects of local forest structure and composition across broad-scale environmental gradients, we were able to better characterize the influence of a tree-killing insect, the WPB, compared to using correlates of tree stress alone.

**Strong positive main effect of CWD.** We found a strong positive effect of site-level CWD on ponderosa pine mortality rate. We did

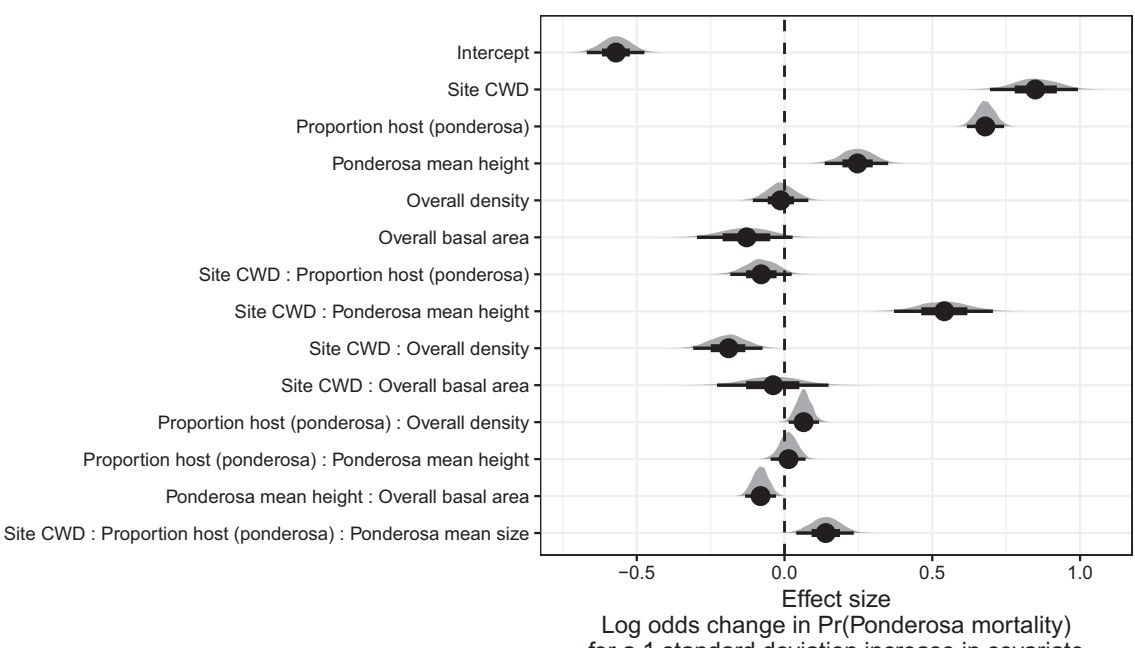

**Fig. 1 Posterior distributions of effect size from zero-inflated binomial model predicting the probability of ponderosa pine mortality in a 20 × 20-m cell given forest structure characteristics and site-level climatic water deficit (CWD).** The gray filled area for each model covariate represents the probability density of the posterior distribution, the point underneath each density curve represents the median of the estimate, the bold interval surrounding the point estimate represents the 66% credible interval, and the thin interval surrounding the point estimate represents the 95% credible interval. Estimates for all model parameters, including Gaussian Process parameters for each site, can be found in Supplementary Table 2.

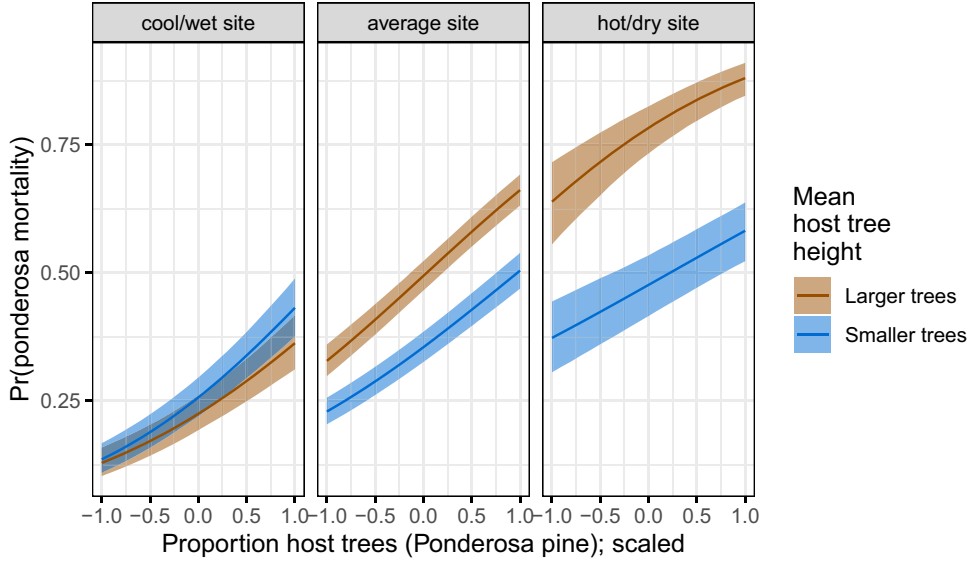

**Fig. 2 Line version of model results with 95% credible intervals showing the primary influence of ponderosa pine structure on the probability of ponderosa pine mortality, and the interaction across climatic water deficit.** The "larger trees" line represents the mean height of ponderosa pine 0.7 standard deviations above the mean (approximately 24.1 m), and the "smaller trees" line represents the mean height of ponderosa pine 0.7 standard deviations below the mean (~12.1 m).

not measure tree water stress at an individual tree level as in other recent work[15], and instead treated CWD as a general indicator of tree stress following results of coarser-scale studies[11]. When measured at a fine scale, even if not at an individual tree level, progressive canopy water loss can be a good indicator of tree water stress and increased vulnerability to mortality from drought or bark beetles[5]. Though our entire study area experienced exceptional hot drought between 2012 and 2016[2,3], using a 30-year historic average of CWD as a site-level indicator of tree stress

does not allow us to disentangle whether water availability was lower in an absolute sense during the drought or whether increasing tree vulnerability to bark beetles was driven by chronic water stress at these historically hotter/drier sites[54].

**Positive effect of host proportion and density.** A number of mechanisms associated with the relative abundance of species in a local area might underlie the strong effect of host proportion on

the probability of host tree mortality. Frequency-dependent herbivory—whereby mixed-species forests experience less herbivory compared to monocultures (as an extreme example)—is common, especially for oligophagous insect species[40]. Non-host volatiles reduce the attraction of several species of bark beetles to their aggregation pheromones[55], including WPB[56]. Combinations of non-host volatiles and an anti-aggregation pheromone have been used successfully to reduce levels of tree mortality attributed to WPB in California[57,58]. The positive relationship between host density and susceptibility to colonization by bark beetles has been so well-documented at the experimental plot level[43,59,60] that lowering stand densities through the selective harvest of hosts is commonly recommended for reducing future levels of tree mortality attributed to bark beetles[61], including WPB[18]. Greater host density shortens the flight distance required for WPB to disperse to new hosts, which likely facilitates bark beetle spread, however, we calibrated our aerial tree detection to ~400 m² areas rather than to individual tree locations, so our data are insufficient to address these relationships. Increased density of ponderosa pine, specifically, may disproportionately increase the competitive environment for host trees (and thus increase their susceptibility to WPB colonization) if intraspecific competition amongst ponderosa pine trees is stronger than the interspecific competition as would be predicted with coexistence theory[62]. Finally, greater host densities increase the frequency that searching WPB land on hosts, rather than non-hosts, thus reducing the amount of energy expended during host finding and selection as well as the time that searching WPB spend exposed to a variety of predators outside the host tree.

**No main effect of overall density, but interaction with CWD.** We detected no relationship between overall tree density and ponderosa pine mortality, though work from the coincident ground plots showed a negative but weak relationship when using proportion of trees killed as a response[14]. Kaiser et al.[28] also show greater mountain pine beetle (*Dendroctonus ponderosae*) infestation in lower-density sites in Montana However, Hayes et al.[31] and Fettig et al.[14] found that measures of overall tree density explained more variation in tree mortality than measures of host availability, though those conclusions were based on broader-scale analyses[31] or a different response variable (i.e., "total number of dead host trees"[14] rather than a binomial response of "number of dead host trees conditional on the total number of host trees" as in our study).

Our greater sample size may have enabled us to more finely parse the role of multi-faceted forest structure and composition, along with CWD and interactions, in driving ponderosa pine mortality rates. Indeed, we did find a negative two-way interaction between site CWD and overall density, suggesting denser stands experienced lower rates of ponderosa mortality in hotter, drier sites, which comports with Restaino et al.[9] in results from their unmanipulated gradient of overall density in the same region during the same hot drought. In the absence of active management, forest structure is largely a product of climate and, with increasing importance at finer spatial scales, topographic conditions[63]. Denser forest patches in our study may indicate greater local water availability, more favorable conditions for tree growth and survivorship, and increased resistance to beetle-induced tree mortality, especially when denser patches are found in hot, dry sites[9,63,64].

**Effect of overall basal area.** While overall tree density is likely an indicator of favorable microsites in fire-suppressed forests, the overall basal area is a better indicator of the local competitive environment, especially in water-limited forests[63,64]. However, we

found no main effect of overall basal area on the probability of ponderosa mortality, nor of its interaction with site-level CWD. This contrasts with the results from Young et al.[11], and from analyses of coincident field plots[14]. While the contrast to Young et al.[11] might be explained by different scales of analyses (i.e., 3500 × 3500 m pixels vs. 20 × 20 m pixels), the contrast with the coincident ground plots is more puzzling. One explanation is that the drone sampling captured more area beyond the conditionally sampled field plots (i.e., 10% ponderosa pine basal area mortality was a criterion for plot selection) that reflected a different relationship between local basal area and tree mortality. Perhaps more likely is that our measure of the total basal area is not precise enough to represent the local competitive environment compared to the field-derived basal area. For our study, the basal area was derived from species-specific and inherently noisy allometric relationships with tree height, which itself was derived from the SfM processing of drone imagery. As remote sensing technology improves to enable finer-scale information extraction (e.g., individual tree measurements), more dialog between ecologists of all stripes[65–67] is needed to fully imagine how to best measure natural phenomena remotely, either by adopting wheels already invented (e.g., plot basal area) or by innovating something brand new.

**Positive main effect of host tree mean size.** The positive main effect of host tree mean size on ponderosa mortality rates tracks the conventional wisdom on the dynamics of WPB in the Sierra Nevada, as well as other primary bark beetles[18]. WPB exhibits a preference for trees 50.8–76.2 cm DBH[68,69], and a positive relationship between host tree size and levels of tree mortality attributed to WPB was reported by Fettig et al.[14] in the coincident field plots as well as in other recent studies[9,15,70]. Larger trees are more nutritious and are therefore ideal targets if local bark beetle density is high enough to successfully initiate mass attack and overwhelm tree defenses, as can occur when many trees are under water stress[7,13,24]. In the recent hot drought, we expected that most trees would be under severe water stress, setting the stage for increasing beetle density, successful mass attacks, and targeting of larger trees. Given that our dead tree height calibration was conservative (accounting for underestimates of drone-derived dead tree heights relative to field-measured trees), it is likely that the positive main effect of tree height that we report represents a lower bounds of this effect. Additionally, Fettig et al.[14] found no tree size/mortality relationship for incense cedar or white fir in the coincident field plots. These species represent 22.3% of the total tree mortality observed in their study, yet in our study, all dead trees were classified as ponderosa pine (see "Methods" section), which could have further dampened the positive effect of tree size on tree mortality that we identified.

**Cross-scale interaction of CWD and host tree size.** In hotter, drier sites, a larger average host size increased the probability of host mortality. Notably, a similar pattern was shown by Stovall et al.[65] in a study confined to the southern Sierra Nevada (i.e., the hottest, driest portion of the more spatially extensive results we present here) with a strong positive tree height/mortality relationship in areas with the greatest vapor pressure deficit and no tree height/mortality relationship in areas with the lowest vapor pressure deficit. Our work suggests that the WPB was cueing into different aspects of forest structure across an environmental gradient in a spatial context in a parallel manner to the temporal context noted by Stovall et al.[65] and Pile et al.[70], who observed that mortality was increasingly driven by larger trees as the hot drought proceeded and became more severe. A temporal signal of bark beetles attacking larger and larger host trees reflects the

positive feedback between forest structure and bark beetle population dynamics as the population phase cycles from endemic to epidemic[13]. This positive feedback leading to eruptive population dynamics is well-documented as a temporal phenomenon, and here we show a similar pattern in a spatial context mediated through site-level CWD.

A key difference from the endemic-to-epidemic positive feedback noted by Boone et al.[13] is that none of our study areas were considered to be in an endemic population phase by typical measures of WPB dynamics[31,33]. WPB dynamics at all sites were considered epidemic, with >5 trees killed per ha (Supplementary Table 1). The cross-scale interaction between broad-scale CWD and local-scale host tree size, even amongst populations all in an epidemic phase, highlights the dramatic implications of the positive feedback for landscape-scale tree mortality. The massive tree mortality in hotter/drier Sierra Nevada forests (lower latitudes and elevations[4,11]) during 2012 to 2016 hot drought likely arose as a synergistic alignment of environmental conditions and local forest structure that allowed WPB to successfully colonize large trees, rapidly increase in population size, and expand. The unexpectedly low mortality in cooler/wetter Sierra Nevada forests compared to model predictions based on coarser-scale forest structure data[11] may result from a different WPB response to local forest structure due to a lack of an alignment with favorable climate conditions and a weaker positive feedback.

**Limitations and future directions**. We have demonstrated that drones can be effective means of collecting forest data at multiple, vastly different spatial scales to investigate a single, multi-scale phenomenon—from meters in between trees, to hundreds of meters of elevation, to hundreds of thousands of meters of latitude. Some limitations remain but can be overcome with further refinements in the use of this tool for forest ecology. Most of these limitations arise from the classification and measurement of standing dead trees, making it imperative to work with field data for calibration and uncertainty reporting.

The greatest limitation in our study arising from classification uncertainty is in the assumption that all dead trees were ponderosa pine, which we estimate from coincident field plots is true ~73.4% of the time. Because the forest structure factors influencing the likelihood of individual tree mortality during the hot drought depended on tree species[15], we cannot rule out that some of the ponderosa pine mortality relationships to forest structure that we observed may be partially explained by those relationships in other species that were misclassified as ponderosa pine using our methods. However, the overall community composition across our study area was similar[14] and we are able to reproduce similar forest structure/mortality patterns in drone-derived data when restricting the scope of analysis to only trees detected in the footprints of the coincident field plots (Supplementary Fig. 2). Thus, we remain confident that the patterns we observed were driven primarily by the dynamic between WPB and ponderosa pine. While spectral information of foliage could help classify living trees to species, the species of standing dead trees were not spectrally distinct. This challenge of classifying standing dead trees to species implies that a conifer forest systems with less bark beetle and tree host diversity, such as mountain pine beetle outbreaks in relative monocultures of naturally occurring lodgepole pine forests in the Intermountain West, should be particularly amenable to the methods presented here even with minimal further refinement because dead trees will almost certainly belong to a single species and have succumbed to colonization by a single bark beetle species. For similar reasons, these methods would also work particularly well if imagery were also captured prior to the mortality event.

Some uncertainty surrounded our ability to detect trees using the geometry of the dense point clouds derived with SfM. The horizontal accuracy (i.e., longitude/latitude position) of the tree detection was better than the vertical accuracy (i.e., height), which may result from a more significant error contribution by the field-based calculations of tree height compared to tree position relative to plot center (Table 1). Height measurements were particularly challenging for standing dead trees because SfM can fail to produce any points representing narrow, needleless treetops in the resulting dense point cloud. Our conservative calibration of drone-measured tree heights to field-measured heights strengthened the main effect of CWD on host mortality in our model and reversed the effect of host tree height. We report that larger host trees increase the probability of host tree mortality, while models using uncalibrated tree heights show that larger trees decrease host mortality rates (see Supplementary Fig. 3 compared to Fig. 1). While our live/dead classification was fairly accurate (96.4% on a withheld data set), our species classifier would likely benefit from better crown segmentation because the pixel-level reflectance values within each crown are averaged to characterize the "spectral signature" of each tree. With better delineation of each tree crown, the mean value of pixels within each tree crown will likely be more representative of that tree's spectral signature.

Better tree detection, crown segmentation, and dead tree height measurement would likely improve with better SfM point clouds which can be enhanced with greater overlap between images[71] or with oblique (i.e., off-nadir) imagery[72]. Frey et al.[71] found that 95% overlap was preferable for generating dense point clouds in forested areas, and James and Robson[72] reduced dense point cloud errors using imagery taken at 30 degrees off-nadir. We only achieved 91.6% overlap with the X3 RGB camera and 83.9% overlap with the multispectral camera, and all imagery was nadir-facing. We anticipate that computer vision and deep learning will also prove helpful in overcoming some of these detection and classification challenges[73].

Finally, we note our study is constrained by the uncertainty in measuring basal area from SfM processing of drone-derived imagery. This uncertainty makes it challenging to represent typical field-based measures of the local competitive environment (e.g., total plot basal area) or ecosystem impact (e.g., the proportion of dead basal area in a plot) in statistical analysis. Instead, we opted to use the probability of ponderosa mortality as our key response variable, which is well-suited to understanding the dynamics between WPB colonization behavior and host tree susceptibility.

**Conclusions**. Climate change adaptation strategies emphasize management action that considers whole-ecosystem responses to inevitable change[74], which requires a macroecological understanding of how phenomena at multiple scales can interact. Tree vulnerability to environmental stressors presents only a partial explanation for tree mortality patterns during hot droughts, especially when bark beetles are present. We have shown that drones can be a valuable tool for investigating multi-scalar phenomena, such as how local forest structure combines with environmental conditions to shape forest insect disturbance. Understanding the conditions that drive dry western U.S. forest responses to disturbances such as bark beetle outbreaks will be vital for predicting outcomes from increasing disturbance frequency and intensity exacerbated by climate change[75]. Our study suggests that outcomes will depend on interactions between local forest structure and broad-scale environmental gradients, with the potential for cross-scale interactions to enhance our understanding of forest insect dynamics.

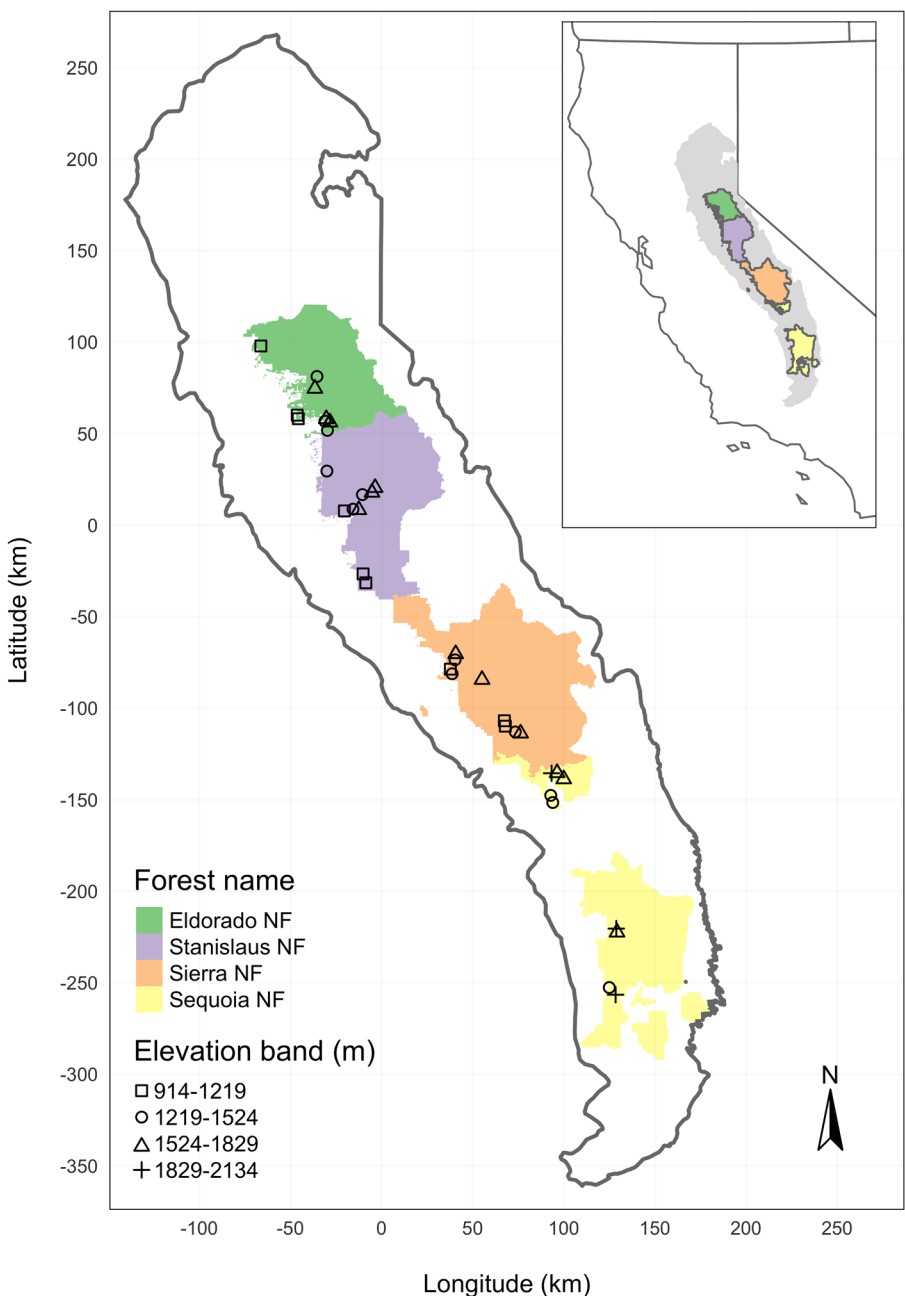

**Fig. 3 The network of field plots spanned a 350-km latitudinal gradient from the Eldorado National Forest in the north to the Sequoia National Forest in the south.** Plots were stratified by three elevation bands in each forest, with the plots in the Sequoia National Forest (the southern-most National Forest) occupying elevation bands 305 m above the three bands in the other National Forests in order to capture a similar community composition.

## Methods

**Study system**. We designed the aerial survey to coincide with 160 vegetation/forest insect monitoring plots at 32 sites established between 2016 and 2017 by Fettig et al.[14] (Fig. 3). The study sites were chosen to reflect typical west-side Sierra Nevada yellow pine/mixed-conifer forests and were dominated by ponderosa pine[14]. Sites were placed in WPB-attacked, yellow pine/mixed-conifer forests across the Eldorado, Stanislaus, Sierra, and Sequoia National Forests and were stratified by elevation (914–1219 m, 1219–1524 m, 1524–1829 m above sea level). In the Sequoia National Forest, the southern-most National Forest in our study, sites were stratified with the lowest elevation band of 1219–1524 m and extended to an upper elevation band of 1829–2134 m to capture a more similar forest community composition as at the more northern National Forests. The sites have variable forest structure and plot locations were selected in areas with >35% ponderosa pine basal area and >10% ponderosa pine mortality. At each site, five 0.041-ha circular plots were installed along transects with 80–200 m between plots. In the field, Fettig et al.[14] mapped all stem locations relative to the center of each plot using azimuth/distance measurements. Tree identity to species, tree height,

and diameter at breast height (DBH) were recorded if DBH was greater than 6.35 cm. The year of mortality was estimated based on needle color and retention if it occurred prior to plot establishment and was directly observed thereafter during annual site visits. A small section of bark (~625 cm$^2$) on both north and south aspects was removed from dead trees to determine if bark beetle galleries were present. The shape, distribution, and orientation of galleries are commonly used to distinguish among bark beetle species[18]. In some cases, deceased bark beetles were present beneath the bark to supplement identifications based on gallery formation. During the spring and early summer of 2018, all field plots were revisited to assess whether dead trees had fallen[14].

In the typical life cycle of WPBs, females initiate host colonization by tunneling through the outer bark and into the phloem and outer xylem where they rupture resin canals. As a result, oleoresin exudes and collects on the bark surface, as is commonly observed with other bark beetle species. During the early stages of the attack, females release an aggregation pheromone component which, in combination with host monoterpenes released from pitch tubes, is attractive to conspecifics[76]. An anti-aggregation pheromone component is produced during the

latter stages of host colonization by several pathways and is thought to reduce intraspecific competition by altering adult behavior to minimize overcrowding of developing brood within the host[77]. Volatiles from several non-hosts sympatric with ponderosa pine have been demonstrated to inhibit the attraction of WPB to its aggregation pheromones[56,78]. In California, WPB generally has 2–3 generations in a single year and can often outcompete other primary bark beetles such as the mountain pine beetle in ponderosa pines, especially in larger trees[33]. WPB population growth rates can, however, be reduced by competition with other beetle species cohabitating in the same host tree, as well as by predation during dispersal to seek a host[33].

**Aerial data collection and processing.** Nadir-facing imagery was captured using a gimbal-stabilized DJI Zenmuse X3 broad-band red/green/blue (RGB) camera[79] and a fixed-mounted Micasense Rededge3 multispectral camera with five narrow bands[80] on a DJI Matrice 100 aircraft[81]. The imagery was captured from both cameras along preprogrammed aerial transects over ~40 ha surrounding each of the 32 sites (each of these containing five field plots) and was processed in a series of steps to yield local forest structure and composition data suitable for our statistical analyses. All images were captured in 2018 during a 3-month period between early April and early July, and thus our work represents a postmortem investigation into the drivers of cumulative tree mortality. Following the call by Wyngaard et al.[82], we establish "data product levels" to reflect the image processing pipeline from raw imagery (Level 0) to calibrated, fine-scale forest structure and composition information on regular grids (Level 4), with each new data level derived from levels below it. Here, we outline the steps in the processing and calibration pipeline visualized in Fig. 4, and include additional details in the Supplementary Methods.

**Level 0: Raw data from sensors.** Raw data comprised ~1900 images per camera lens (one broad-band RGB lens and five narrow-band multispectral lenses) for each of the 32 sites (Fig. 4; Level 0; Supplementary Figs. 4 and 5). Prior to the aerial survey, two strips of bright orange drop cloth (~100 ×15 cm) were positioned as an "X" over the permanent monuments marking the center of the 5 field plots from Fettig et al.[14] (Supplementary Fig. 6).

We preprogrammed north-south aerial transects using Map Pilot for DJI on iOS flight software[84] at an altitude of 120 m above ground level (with "ground" defined using a 1-arc-second digital elevation model[85]). The resulting ground sampling distance was ~5 cm/px for the Zenmuse X3 RGB camera and ~8 cm/px for the Rededge3 multispectral camera. We used 91.6% image overlap (both forward and side) at the ground for the Zenmuse X3 RGB camera and 83.9% overlap (forward and side) for the Rededge3 multispectral camera.

**Level 1: Basic outputs from photogrammetric processing.** We used SfM photogrammetry implemented in Pix4Dmapper Cloud (www.pix4d.com) to generate dense point clouds (Fig. 4; Level 1, left; Supplementary Fig. 7), orthomosaics (Fig. 4; Level 1, center and Supplementary Fig. 8), and digital surface models (Fig. 4; Level 1, right and Supplementary Fig. 9) for each field site[71]. For 29 sites, we processed the Rededge3 multispectral imagery alone to generate these products. For three sites, we processed the RGB and the multispectral imagery together to enhance the point density of the dense point cloud. All SfM projects resulted in a single processing "block," indicating that all images in the project were optimized and processed together. The dense point cloud represents $x$, $y$, and $z$ coordinates as well as the color of millions of points per site. The orthomosaic represents a radiometrically uncalibrated, top-down view of the survey site that preserves the relative $x$–$y$ positions of objects in the scene. The digital surface model is a rasterized version of the dense point cloud that shows the altitude above sea level for each pixel in the scene at the ground sampling distance of the camera that generated the Level 0 data.

**Level 2: Corrected outputs from photogrammetric processing.**
*Radiometric corrections.* A radiometrically corrected reflectance map (Fig. 4; Level 2, left two figures; i.e., a corrected version of the Level 1 orthomosaic; Supplementary Fig. 10) was generated using the Pix4D software by incorporating incoming light conditions for each narrow band of the Rededge3 camera (captured simultaneously with the Rededge3 camera using an integrated downwelling light sensor) as well as a pre-flight image of a calibration panel of known reflectance (see Supplementary Table 3 for camera and calibration panel details).

*Geometric corrections.* We implemented a geometric correction to the Level 1 dense point cloud and digital surface model by normalizing these data for the terrain underneath the vegetation. We generated the digital terrain model representing the ground underneath the vegetation at 1-m resolution (Fig. 4; Level 2, third image and Supplementary Fig. 11) by classifying each survey area's dense point cloud into "ground" and "non-ground" points using a cloth simulation filter algorithm[86] implemented in the lidR[53] package and rasterizing the ground points using the raster package[87]. We generated a canopy height model (Fig. 4; Level 2, fourth image and Supplementary Fig. 12) by subtracting the digital terrain model from the digital surface model.

## Level 3: Domain-specific information extraction
*Level 3a: Data derived from spectral or geometric Level 2 product.* Using just the spectral information from the radiometrically corrected reflectance maps, we calculated several vegetation indices including the normalized difference vegetation index[83] (NDVI; Fig. 4; Level 3a, first image and Supplementary Fig. 13), the normalized difference red edge[88] (NDRE), the red-green index[89] (RGI), the red edge chlorophyll index[90] ($CI_{red\ edge}$), and the green chlorophyll index[90] ($CI_{green}$).

Using just the geometric information from the canopy height model or terrain-normalized dense point cloud, we generated maps of detected trees (Fig. 4; Level 3a, second and third images and Supplementary Fig. 14) by testing a total of 7 automatic tree detection algorithms and a total of 177 parameter sets (Table 2). We used the field plot data to assess each tree detection algorithm/parameter set by converting the distance-from-center and azimuth measurements of the trees in the field plots to $x$–$y$ positions relative to the field plot centers distinguishable in the Level 2 reflectance maps as the orange fabric X's that we laid out prior to each flight. In the reflectance maps, we located 110 out of 160 field plot centers while some plot centers were obscured due to dense interlocking tree crowns or because a plot center was located directly under a single tree crown. For each of the 110 field plots with identifiable plot centers– the "validation field plots", we calculated 7 forest structure metrics using the ground data collected by Fettig et al.[14]: total number of trees, number of trees greater than 15 m in height, mean height of trees, 25th percentile tree height, 75th percentile tree height, mean distance to nearest tree neighbor, and mean distance to second nearest neighbor. For each tree detection algorithm and parameter set described above, we calculated the same set of 7 structure metrics within the footprint of the validation field plots. We calculated the Pearson's correlation and root mean square error (RMSE) between the ground data and the aerial data for each of the 7 structure metrics for each of the 177 automatic tree detection algorithms/parameter sets. For each algorithm and parameter set, we calculated its performance relative to other algorithms as to whether its Pearson's correlation was within 5% of the highest Pearson's correlation as well as whether its RMSE was within 5% of the lowest RMSE. We summed the number of forest structure metrics for which it reached these 5% thresholds for each algorithm/parameter set. For automatically detecting trees across the whole study, we selected the algorithm/parameter set that performed well across the most forest metrics (see "Results" section).

We delineated individual tree crowns (Fig. 4; Level 3a, fourth image and Supplementary Fig. 15) with a marker controlled watershed segmentation algorithm[99] implemented in the ForestTools package[97] using the detected treetops as markers. If the automatic segmentation algorithm failed to generate a crown segment for a detected tree (e.g., often snags with a very small crown footprint), a circular crown was generated with a radius of 0.5 m. If the segmentation generated multiple polygons for a single detected tree, only the polygon containing the detected tree was retained. Because image overlap decreases near the edges of the overall flight path and reduces the quality of the SfM processing in those areas, we excluded segmented crowns within 35 m of the edge of the survey area. Given the narrower field of view of the Rededge3 multispectral camera versus the X3 RGB camera whose optical parameters were used to define the ~40 ha survey area around each site, as well as the 35 m additional buffering, the survey area at each site was ~30 ha (Supplementary Table 1).

*Level 3b: Data derived from spectral and geometric information.* We overlaid the segmented crowns on the reflectance maps from 20 sites spanning the latitudinal and elevation gradient in the study. Using QGIS (https://qgis.org/en/site/), we hand classified 564 trees as live/dead and as one of 5 dominant species in the study area (ponderosa pine, *Pinus lambertiana*, *Abies concolor*, *Calocedrus decurrens*, or *Quercus kelloggi*) using the mapped ground data as a guide. Each tree was further classified as "host" for ponderosa pine or "non-host" for all other species[18]. We extracted all the pixel values within each segmented crown polygon from the five, Level 2 orthorectified reflectance maps (one per narrow band on the Rededge3 camera) as well as from the five, Level 3a vegetation index maps using the velox package[100]. For each crown polygon, we calculated the mean value of the extracted Level 2 and Level 3a pixels and used them as ten independent variables in a five-fold cross-validated boosted logistic regression model to predict whether the hand classified trees were alive or dead. For just the living trees, we similarly used all 10 mean reflectance values per crown polygon to predict tree species using a five-fold cross-validated regularized discriminant analysis. The boosted logistic regression and regularized discriminant analysis were implemented using the caret package in R[101]. We used these models to classify all tree crowns in the data set as alive or dead (Fig. 4; Level 3b, first image and Supplementary Fig. 16) as well as to classify the species of living trees (and then host or non-host; Fig. 4; Level 3b, second image; Supplementary Fig. 17).

Because the tops of dead, needleless trees are narrow, they may not be well-represented in the point clouds produced using SfM photogrammetry, which biases their height estimates downward. Further, field measurements can overestimate the heights of live trees relative to aerial survey methods[102]. To correct these measurement biases, we calibrated aerial tree height measurements to ground-based height measurements. Specifically, we identified the crowns of 451 field-measured trees in the drone-derived tree data, modeled the relationship between field- and drone-measured tree heights for both live and dead trees, and used the models to adjust the drone-measured tree heights (Supplementary Methods). We applied a conservative height correction to live and dead trees based on trees measured by the

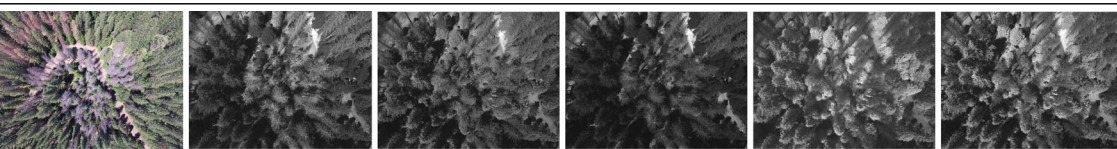

Level 0: raw data from sensors

Level 1: basic outputs from photogrammetric processing

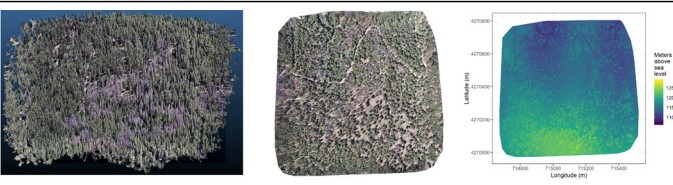

Level 2: corrected outputs from photogrammetric processing

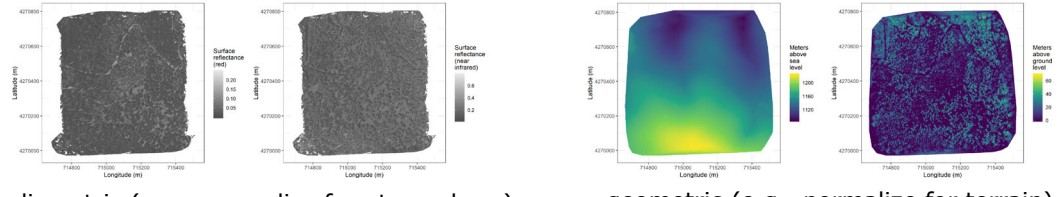

radiometric (e.g., normalize for atmosphere)     geometric (e.g., normalize for terrain)

Level 3: domain-specific information extraction

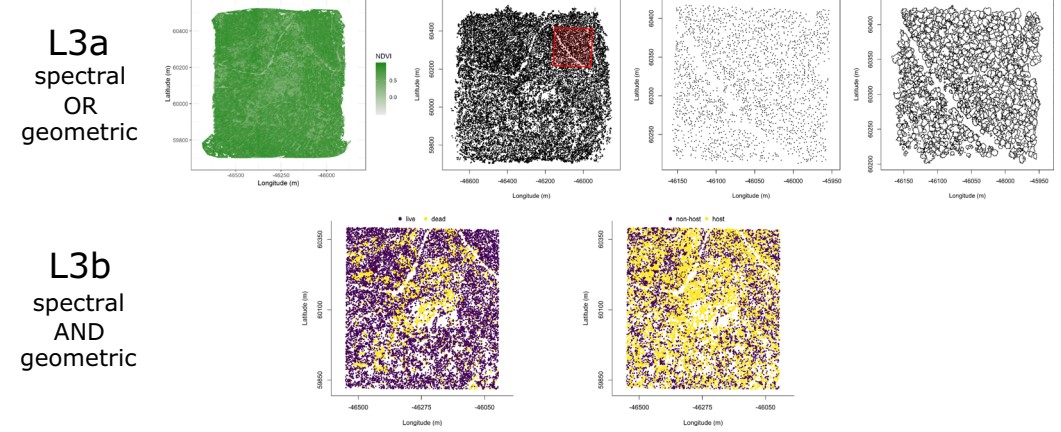

L3a
spectral
OR
geometric

L3b
spectral
AND
geometric

Level 4: aggregations to regular grids

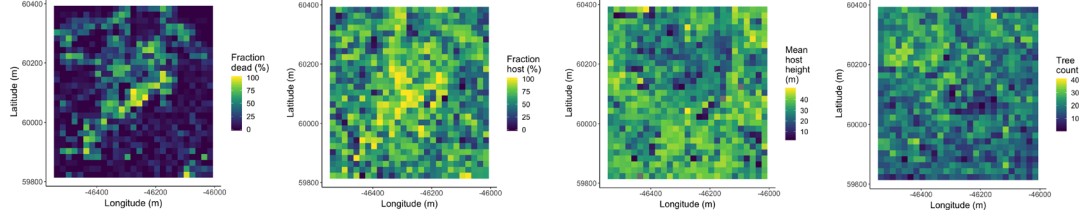

drone to be greater than 20 m in height that increased dead tree height by an average of 2.8 m and reduced the heights of live trees by an average of 0.9 m (Supplementary Figs. 18–20 and Supplementary Note 2). Finally, we estimated the basal area of each tree from their corrected drone-measured height using species-specific simple linear regressions of the relationship between height and DBH as measured in the coincident field plots from Fettig et al.[14]

We note that our study relies on the generation of Level 3a products in order to combine them and create Level 3b products like the classified tree maps, but this need not be the case. For instance, deep learning/neural net methods may be able to use both the spectral and geometric information from lower-level data products simultaneously to locate and classify trees in a scene and directly generate Level 3b products without a need to first generate the Level 3a products[103,104].

**Level 4: Aggregations to regular grids.** We rasterized the forest structure and composition data at a spatial resolution similar to that of the field plots to better

**Fig. 4 Schematic of the data processing workflow for a single site with each new data product level derived from data at lower levels.** Level 0 represents raw data from the sensors. From left to right: RGB photo from DJI Zenmuse X3, output images from Micasense Rededge3 (blue, green, red, near infrared, red edge). Level 1 represents basic outputs from the SfM workflow. From left to right: dense point cloud, RGB orthomosaic, digital surface model (DSM; ground elevation plus vegetation height). Level 2 represents radiometrically or geometrically corrected Level 1 products. From left to right: radiometrically corrected "red" surface reflectance map, radiometrically corrected "near infrared" surface reflectance map, digital terrain model (DTM) derived by a geometric correction of the dense point cloud, canopy height model (CHM; DTM subtracted from the DSM). Level 3 represents domain-specific information extraction from Level 2 products and is divided into two sub-levels. Level 3a products are derived using only spectral or only geometric data. From left to right: map of Normalized Difference Vegetation Index (NDVI)[83], map of detected trees derived from the CHM, detected trees within the red polygon, polygons representing segmented tree crowns within a red polygon. Level 3b products are derived using both spectral and geometric data. From left to right: trees classified as alive or dead based on spectral reflectance within each segmented tree crown, trees classified as WPB host/non-host. Level 4 represents aggregations of Level 3 products to regular grids that better reflects the grain size of the validation (e.g., to match the area of validation field plots) or which provides neighborhood- rather than individual-scale information (e.g., stand-level proportion of host trees). From left to right: grid representing a fraction of dead trees per cell, grid representing a fraction of hosts per cell, grid representing mean host height per cell, tree density per cell. All cells measure 20 × 20 m.

**Table 2 Algorithm name, number of parameter sets tested for each algorithm, and references.**

| Algorithm | Parameter sets tested | Reference(s) |
|---|---|---|
| li2012 | 131 | Li et al.[91]; Jakubowski et al.[92]; Shin et al.[93] |
| lmfx | 30 | Roussel[94] |
| localMaxima | 6 | Roussel et al.[53] |
| multichm | 1 | Eysn et al.[95] |
| ptrees | 3 | Vega et al.[96] |
| vwf | 3 | Plowright[97] |
| watershed | 3 | Pau et al.[98] |

match the grain size at which we validated the automatic tree detection algorithms. In each raster cell, we calculated: number of dead trees, number of ponderosa pine trees, total number of trees, and mean height of ponderosa pine trees. The values of these variables in each grid cell and derivatives from them were used for visualization and modeling. Here, we show the fraction of dead trees per cell (Fig. 4; Level 4, first image and Supplementary Fig. 21), the fraction of host trees per cell (Fig. 4; Level 4, second image), the mean height of ponderosa pine trees in each cell (Fig. 4; Level 4, third image), and the total count of trees per cell (Fig. 4; Level 4, fourth image).

**Note on assumptions about dead trees**. For the purposes of this study, we assumed that all dead trees were ponderosa pine and thus hosts colonized by WPB. This is a reasonably good assumption for our study area; for example, Fettig et al.[14] found that 73.4% of dead trees in their coincident field plots were ponderosa pine. Mortality was concentrated in the larger-diameter classes and attributed primarily to WPB (see Fig. 5 of Fettig et al.[14]). The species contributing to the next highest proportion of dead trees was incense cedar which represented 18.72% of the dead trees in the field plots. While the detected mortality is most likely to be ponderosa pine killed by WPB, it is critical to interpret our results with these limitations in mind.

**Environmental data**. We used CWD[105] from the 1981–2010 mean value of the basin characterization model[106] as an integrated measure of historic temperature and moisture conditions for each of the 32 sites. Higher values of CWD correspond to historically hotter, drier conditions and lower values correspond to historically cooler, wetter conditions. CWD has been shown to correlate well with broad patterns of tree mortality in the Sierra Nevada[11] as well as bark beetle-induced tree mortality[107]. The forests along the entire CWD gradient used in this study experienced exceptional hot drought between 2012 and 2016 with severity of at least a 1200-year event, and perhaps more severe than a 10,000-year event[2,3]. We converted the CWD value for each site into a z-score representing that site's deviation from the mean CWD across the climatic range of Sierra Nevada ponderosa pine as determined from 179 herbarium records described in Baldwin et al.[108]. Thus, a CWD z-score of 1 would indicate that the CWD at that site is one standard deviation hotter/drier than the mean CWD across all geolocated herbarium records for ponderosa pine in the Sierra Nevada.

**Statistical model**. We used a generalized linear model with a zero-inflated binomial response and a logit link to predict the probability of ponderosa pine mortality within each 20 × 20-m cell using the total number of ponderosa pine trees in each cell as the number of trials, and the number of dead trees in each cell as the number of "successes". As covariates, we used the proportion of trees that are WPB hosts (i.e., ponderosa pine) in each cell, the mean height of ponderosa pine trees in each cell, the count of trees of all species (overall density) in each cell, and the site-level CWD using Eq. 1. Note that the two-way interaction between the overall density and the proportion of trees that are hosts is directly proportional to the number of ponderosa pine trees in the cell. We centered and scaled all predictor values, and used weaklyregularizing default priors from the brms package[109]. To measure and account for spatial autocorrelation underlying ponderosa pine mortality, we sub-sampled the data at each site to a random selection of 200, 20 × 20-m cells representing ~27.5% of the surveyed area. Additionally, with these subsampled data, we included a separate exact Gaussian process term per site of the non-centered/nonscaled interaction between the x- and y-position of each cell using the gp() function in the brms package[109]. The Gaussian process estimates the spatial covariance in the response variable (log-odds of ponderosa pine mortality) jointly with the effects of the other covariates.

$$
y_{i,j} \sim \begin{cases} 0, & p \\ Binom(n_i, \pi_i), & 1-p \end{cases}
$$
$$
\begin{aligned}
\log it(\pi_i) = \beta_0 + \\
\beta_1 X_{cwd,j} + \beta_2 X_{propHost,i} + \beta_3 X_{PipoHeight,i} + \\
\beta_4 X_{overallDensity,i} + \beta_5 X_{overallBA,i} + \\
\beta_6 X_{cwd,j} X_{PipoHeight,i} + \beta_7 X_{cwd,j} X_{propHost,i} + \\
\beta_8 X_{cwd,j} X_{overallDensity,i} + \beta_9 X_{cwd,j} X_{overallBA,i} + \\
\beta_{10} X_{propHost,i} X_{PipoHeight,i} + \beta_{11} X_{propHost,i} X_{overallDensity,i} + \\
\beta_{12} X_{PipoHeight,i} X_{overallBA,i} + \\
\beta_{13} X_{cwd,j} X_{propHost,i} X_{PipoHeight,i} + \\
\mathcal{GP}_j(x_i, y_i)
\end{aligned}
\tag{1}
$$

Where $y_i$ is the number of dead trees in cell $i$, $n_i$ is the sum of the dead trees (assumed to be ponderosa pine) and live ponderosa pine trees in cell $i$, $\pi_i$ is the probability of ponderosa pine tree mortality in cell $i$, $p$ is the probability of there being zero dead trees in a cell arising as a result of an independent, unmodeled process, $X_{cwd,j}$ is the z-score of CWD for site $j$, $X_{propHost,i}$ is the scaled proportion of trees that are ponderosa pine in cell $i$, $X_{PipoHeight,i}$ is the scaled mean height of ponderosa pine trees in cell $i$, $X_{overallDensity,i}$ is the scaled density of all trees in cell $i$, $X_{overallBA,i}$ is the scaled basal area of all trees in cell $i$, $x_i$ and $y_i$ are the x- and y-coordinates of the centroid of the cell in an EPSG3310 coordinate reference system, and GPj represents the exact Gaussian process describing the spatial covariance between cells at site $j$.

We fit this model using the brms package[109] which implements the No U-Turn Sampler extension to the Hamiltonian Monte Carlo algorithm[110] in the Stan programming language[111]. We used 4 chains with 5000 iterations each (2000 warmup, 3000 samples), and confirmed chain convergence by ensuring all Rhat values were less than 1.1[112] and that the bulk and tail effective sample sizes (ESS) for each estimated parameter were greater than 100 times the number of chains (i.e., >400 in our case). We used posterior predictive checks to visually confirm model performance by overlaying the density curves of the predicted number of dead trees per cell over the observed number[113]. For the posterior predictive checks, we used 50 random samples from the model fit to generate 50 density curves and ensured curves were centered on the observed distribution, paying special attention to model performance at capturing counts of zero (Supplementary Fig. 22).

## Data availability

All field and drone data processed for this study are available via the Open Science Framework at https://doi.org/10.17605/OSF.IO/3CWF9[114]. The administrative boundaries file for the USDA Forest Service (S_USA.AdministrativeForest.shp) can be found at https://data.fs.usda.gov/geodata/edw/datasets.php?dsetCategory=boundaries. The 2014 version of the 1981–2010 thirty-year historic average climatic water deficit data (cwd1981_2010_ave_HST_1550861123.tif) can be found on the California Climate Commons at http://climate.calcommons.org/dataset/2014-CA-BCM. The data set representing ponderosa pine geolocations derived from herbaria records (California_Species_clean_All_epsg_3310.csv) can be found at https://doi.org/10.6078/D16K5W[115]. The vector file representing Jepson geographic subdivisions of California and used to define the Sierra Nevada region can be requested at https://ucjeps.berkeley.edu/eflora/geography.html.

## Code availability

Statistical analyses were performed using the brms packages. With the exception of the SfM software (Pix4Dmapper Cloud) and the GIS software QGIS, all data carpentry and analyses were performed using R[116]. All code used to generate the results from this study are available via GitHub at and is mirrored on the Open Science Framework at https://doi.org/10.17605/OSF.IO/WPK5Z[117].

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

## Acknowledgements

We gratefully acknowledge funding from the USDA Forest Service Western Wildlands Environmental Threat Assessment Center (WWETAC) and the Pacific Southwest Research Station Climate Change Competitive Grant Program. We thank Connie Millar for comments and guidance during the development of this project, and Meagan Oldfather for her role as visual observer during drone flights. We thank Beverly Bulaon for her significant contribution to field plot network establishment. We also thank Victoria Scholl for helpful discussions regarding remotely-sensed data product levels, and Derek Young for helpful discussions while revising this manuscript. We gratefully acknowledge the Map Pilot for iOS team (who implemented several feature requests that helped us conduct our drone flights), Pix4D (which provided free cloud infrastructure for much of the Structure from Motion photogrammetry processing), and the Open Science Framework (who facilitated the public access to our complete dataset). Thanks to Alex Mandel, Dan Krofcheck, Taylor Nelson, Nate Metzler, Brandon Stark, Andy Wong, Grace Liu, Sean Hogan, the Micasense team, Lawrence Dennis from Aerial Technology International, and Casey Neistat for valuable input regarding drones, sensors, safe flying, and SfM photogrammetry. Publication of this article was partially funded by the University of Colorado Boulder Libraries Open Access Fund and the University of California, Davis Library Open Access Fund.

## Author contributions

Author contributions are defined using the Contributor Roles Taxonomy (CRediT; https://casrai.org/credit/). Conceptualization: M.J.K., A.M.L., C.J.F., M.P.N., and L.A.M.; data curation: M.J.K.; formal analysis: M.J.K.; funding acquisition: M.J.K., M.P.N., C.J.F., and A.M.L.; investigation: M.J.K., L.A.M., and C.J.F.; methodology: M.J.K. and A.M.L.; project administration: M.J.K., M.P.N., and A.M.L.; resources: M.J.K., M.P.N., and A.M.L.; software: M.J.K.; supervision: M.J.K., M.P.N., and A.M.L.; validation: M.J.K.; visualization: M.J.K.; writing—original draft: M.J.K.; writing—review and editing: M.J.K., A.M.L., C.J.F., M.P.N., and L.A.M.

## Competing interests

The authors declare no competing interests.
