## [Peer Review File · Nature Communications]

Reviewer comments, first round –

Reviewer #1 (Remarks to the Author):

Review for NCOMMS-20-01457-T

I appreciated reading the manuscript *Cross-scale interaction of host tree size and climate governs bark beetle-induced tree mortality*, in which the authors present an analysis of forest structural drivers on bark beetle induced tree mortality in a drought-affected region.

The manuscript is very well structured and written, it was a real pleasure reading it. I also appreciate the rigor and transparency of the statistical analysis, making this a valuable piece of science. That said, I have some remarks regarding the overall aim of the manuscript and some minor methodological issues.

1) The ultimate aim of the manuscript is not clear to me. Is the study presenting a novel technique (i.e., using drones) for deriving forest structural attributes that can be used for modelling, or is the focus of the study on deriving novel insights into bark beetle dynamics? The manuscript is very technical and presents a lot of technical details. The main conclusion also is that drones can be “a valuable tool for investigating multi-scalar phenomena...” (L. 533). I hence suspect that the main aim is presenting a novel methodological advancement, rather than generating exiting new insights into bark beetle dynamics. I encourage the authors to be crystal clear on the aim of their study.

2) The authors state that using drones can enhance the set of predictors commonly used for predicting bark beetle outbreak dynamics by moving from stand-based to tree-based variables (L. 88). In their analysis, however, they only use very generic indicators (tree height, proportion of ponderosa pine and tree density), which have been investigated in previous studies already. They also aggregated tree-based measures to a common grid (level 4), which is somehow similar to doing analysis at the stand or plot scale (even though a more localized definition of stands). I thus have trouble seeing where the drone-based data offers advantage over more traditional approaches, e.g., using Lidar-based canopy height models or plot-based field measures of tree height, density, species, etc.

3) The questions asked in the introduction are good, but many of them could have been answered based on previous research results. The effect of proportion of host trees and tree density has been shown in many previous studies located in the US, Canada and Europe. Likewise, the positive link between drought and bark beetle activity is also well established in the scientific literature. Novel insights emerge from the interaction of drivers, but the general idea of cross-scale interactions has been established and tested in previous studies (i.e., Raffa et al. 2008, Seidl et al. 2016, Senf et al. 2017).

Overall, my critique on very high level, and I want to emphasize that I really enjoyed reading the paper. I think this is an interesting and robust study that deserves publication. However, I think that the paper would be better suited for a more specialized journal, focusing either on the methodological advances or on the ecological insights generated.

Some minor comments:

L. 53: Why are “primary” and “mass attack” in quotation marks?

Materials and Methods: To my knowledge, the Nature-family doesn't allow figures in the M&M section.

Figure 2: Great figure, it really helps understanding what you do!

L. 313ff: What is the time frame of the analysis? Is the CWD data annual or are you using averages? If averages, what is the time frame the averages were calculated for? Is it 1981-2010? If so, how does this relate to the current drought? Some more information needed here.

L. 337ff: Why do you assume that the zeros emerge from another process? Have you tested the model without zero-inflation term? Ecologically, it could be the same predictors that explain the absence of infestation (i.e., no host trees). Moreover, I didn't see where you report results for parameter p , so the proportion of plots experiencing no mortality. Maybe I missed it, but this would be interesting to see.

L. 345: I appreciate that you used Bayesian statistics and Stan to do so. That kind of models really show the power of Stan and Bayesian stats in general! However, I miss some crucial information on the MCMC sampling. Please note that you used MCMC samplers implemented in Stan and cite Carpenter et al. 2017. Second, you do not mention any priors, even though this is a crucial part of Bayesian analysis. Even if you use weakly informative priors as implemented in brms, report those in the paper as this is essential for reproducibility. Finally, I really appreciate you did posterior predictive checks, but I would love to see the graphs in the supplement.

L. 386: Throughout the discussion, you always start a paragraph by repeating the results. This is very repetitive, and I suggest using each paragraph's topic sentence for highlighting the main implications of the results, rather than repeating what was reported in the results section already.

Literature cited

Carpenter, B. et al. Stan: A Probabilistic Programming Language. *Journal of Statistical Software* 76, 1–32 (2017).

Raffa, K. F. et al. Cross-scale Drivers of Natural Disturbances Prone to Anthropogenic Amplification: The Dynamics of Bark Beetle Eruptions. *BioScience* 58, 501–517 (2008).

Seidl, R. et al. Small beetle, large-scale drivers: how regional and landscape factors affect outbreaks of the European spruce bark beetle. *J Appl Ecol* 53, 530–540 (2016).

Senf, C., Campbell, E. M., Pflugmacher, D., Wulder, M. A. & Hostert, P. A multi-scale analysis of western spruce budworm outbreak dynamics. *Landscape Ecology* 32, 501–514 (2017).

Reviewer #2 (Remarks to the Author):

"Cross-scale interaction ... " addresses an important concern, increased losses of forest resources to native insects whose populations are exacerbated by a changing environment. The study focuses on mortality of ponderosa pine to western pine beetle in California USA, but has implications to bark beetles worldwide.

A major strength of the paper is its emphasis on cross-scale interactions, which are highly difficult to partition despite their widely recognized importance. A strength of the experimental approach is the use of drones to provide spatially explicit data. This approach also brings a significant weakness, namely that all dead trees were assumed to be ponderosa pine, and all of these were assumed to have been killed by western pine beetle. However, the authors recognize this limitation and allot it ample discussion. If it could also be included in the Abstract that would be better yet. The paper is well written. The statistical analyses are appropriate.

Despite this paper's strengths it requires some improvement before it can be acceptable for publication in *Nature Communications*. I identify several issues below, each of which I feel can be achieved in a revision.

Major issues

1. The title refers to 'Climate', but the paper is about Climatic Water Deficit. Climate is a much more expansive term, including temperature (which also affects the beetles directly not just via the host), wind, etc., each of which influences specific features of bark beetle population dynamics. So 'Climate' should be replaced with either 'Climatic Water Deficit' or 'Drought' in the title. This is a simple, but important fix.
2. The second problem is more systemic, and will require some substantial reframing. Specifically, the literature already provides mechanistic explanations for some of the trends reported here, so the appearance of these trends is less surprising than suggested, although I agree they go against conventional dogma. These mechanistic studies should be better integrated into the story.
 - a. For example, the contrasting relationships that host susceptibility and host brood quality have with tree size, and how beetle choice transitions with its population numbers, are mechanistically delineated in Boone et al. 2011. This is also described much more mechanistically and at finer-scale resolution in Raffa et al. 2008, 2015 than depicted here.
 - b. Likewise, the positive effect of host density on beetle attack success can at least partly be attributed to their switching behavior whereby aggregated beetles move to adjacent trees. This is described by Geizler & Gara 1978, Mitchell & Preisler 1991 and Preisler 1993, which should be cited. Also empirical evidence for tree density decreasing defense capacity is described in fir by Raffa & Berryman 1982.
 - c. Important mid-scale data on relationships between moisture availability and bark beetle attack are in Kaiser et al. 2012. These should be cited.

Moderate issues

1. Line 56: 'This Allee effect creates a strong coupling between beetle selection behavior of host trees and host tree susceptibility to colonization' Perhaps it would be better to state 'This Allee effect creates a strong coupling among beetle selection behavior of host trees, host tree susceptibility to colonization, and beetle stand-level density'. Include Wallin & Raffa 2004, as that provides empirical data for how bark beetles change host selection behavior with population density.
2. Line 57: Resin physically expels, or delays, beetles but it also contains chemicals that are toxic to the beetles and their fungi.
3. Line 59, 66, 77, 475: Some examples where Boone et al. 2011 seems particularly relevant. Line 88-91: Kaiser et al. 2012 seems particularly relevant here.
4. Line 146-7. It would be more accurate to state nonhost volatiles inhibited attraction of WPB to its aggregation pheromone, as is stated in lines 408-9.
5. Line 149-50: Should something be mentioned here about other competing bark beetle species besides mountain pine beetle?
6. Fig. 2: I really like the extensive figure caption you provide here!
7. Lines 420-425. The positive relationships you describe between tree growth and tree resistance seem important and perhaps merit stronger emphasis.

Reviewer #3 (Remarks to the Author):

The aims of the study were to develop methods for estimating tree mortality using drone photographs across a broad gradient of ponderosa and mixed conifer forests in the Sierra Nevada Range and determine if local tree height, number of trees and estimated long-term site water availability could be related to tree mortality. The methods developed showed that plot characteristics and tree mortality could be modestly estimated using the drone imagery and post-processing steps, compared to the 110 ~400 m² ground-measured plots visible in the drone imagery. Correlations with plot characteristics and drone imagery ranged from 0.16 for 25th percentile height to 0.67 for total tree height. No comparison for drone-based estimates of mortality to the ground-measured plots was given, though the study claimed 98% success in estimating mortality in a data set withheld from developing the models.

Using the drone data set of plot characteristics, and an areal description of estimated long-term site water availability, models estimated that the proportion of ponderosa pine host trees had the largest effect (positive) on mortality, followed by the intercept and ponderosa mean height (both negative effects), and the combination of ponderosa mean height and estimated long-term site

water availability (positive effect). Tree density and the combination of tree density and estimated long-term site water availability had small negative effects.

Overall, the study suggests that drone imagery can be helpful in identifying some of the stand characteristics that relate to local variability in bark-beetle mortality, but the results are puzzling compared to earlier work (Waring and Pitman 1985), and there are some fairly major issues that need to be addressed.

1. Tree density (used in this paper) is a very poor surrogate for the influence of surrounding trees on the resources of a target individual. Plot basal area is a much better surrogate for the overall influence of surrounding trees on the trees in a plot, because basal area is closely related to both leaf area and biomass—which are both closely related to resource use. The number of trees can be related to resource impact, but only if there are an extraordinary number of very small trees or if there are a large number of large trees. Perhaps this is why the results for 'density' and tree height run counter to earlier studies (Waring and Pitman 1985). Basal area needs to be used!

2. Proportion of tree mortality relative to total trees in a plot is also a very poor surrogate for the effects of that mortality on the ecosystem. It is not the number of trees killed or the relative number of trees that are killed that is important to ecosystem effects such as water use, streamflow, or carbon storage, it is the proportion of basal area that is lost. The use of proportional number of trees killed instead of the proportion of basal area killed may also have contributed to the odd results (tree density and mean height having a negative effect on mortality fraction), and compared to earlier studies. Again, basal area would be much more biologically meaningful than tree numbers.

3. Data for the model estimates of mortality needs to be presented for the ground-measured plots, and the method for estimating mortality for the models needs to be presented.

Finally, 'CWD' is a poor choice of acronym for a study on forests, where it usually means 'Coarse Woody Debris' or 'Chronic Wasting Disease'.

Waring RH, Pitman GB (1985) Modifying lodgepole pine stands to change susceptibility to mountain pine beetle attack. *Ecology* 66:889-897.

Mike Ryan

NCOMMS-20-01457A: Response to reviews

Dear Dr. Mike Ryan, and the two anonymous reviewers,

Thank you very much for your feedback. We have substantially revised our manuscript in response to your comments. Among our revisions are two major changes.

- First, we reframed our introduction to clarify that the primary purpose of the study is to investigate patterns of tree mortality attributed to the western pine beetle in California's 2012-2015 hot drought. We clarified that the methodological innovation was integrating data with both a fine grain size (i.e., individual trees classified as live/dead and with live trees classified to species) and with a broad spatial extent (350 km of latitude and 1000 m of elevation across a strong climatic water deficit gradient). We explain how the methodological development of a forest ecology workflow using drone-derived imagery, while secondary to our primary ecology question, enabled us to identify and partition the cross-scale drivers of the western pine beetle dynamics.*
- Second, we refactored our primary analysis to include basal area as a better correlate of the competitive environment faced by ponderosa pine trees in the study area. Because of the extensive sampling in our study design (both at local and broad scales) that sufficiently decoupled basal area and tree density, we were able to do this while still simultaneously considering forest structure measures that may not reflect competition well, but that still play important roles in western pine beetle dynamics (e.g., overall tree density may not reflect the competitive environment, but is still important given its impact on aggregation/anti-aggregation pheromone plumes).*

Below, we address your comments point-by-point. The exact text from the reviewer comments are copied in this document and are displayed in black italic font and have each been assigned a number for reference (R1.1 for the first point by reviewer one, R2.1 for the first point by reviewer 2, etc.). Our responses to these individual points are written in blue text, and any pertinent direct quotes from our revised manuscript that reflect changes addressing editor/reviewer comments are written in green text. We have also included a Word document of this submitted version with changes tracked from our previous submission.

Thank you very much for your time and consideration.

*Sincerely,
The authors*

RESPONSES TO REVIEWER #1

R1.1: I appreciated reading the manuscript Cross-scale interaction of host tree size and climate governs bark beetle-induced tree mortality, in which the authors present an analysis of forest structural drivers on bark beetle induced tree mortality in a drought-affected region. The manuscript is very well structured and written, it was a real pleasure reading it. I also appreciate the rigor and transparency of the statistical analysis, making this a valuable piece of science. That said, I have some remarks regarding the overall aim of the manuscript and some minor methodological issues.

Thank you very much for the kind words, and for your comments.

R1.2: 1) *The ultimate aim of the manuscript is not clear to me. Is the study presenting a novel technique (i.e., using drones) for deriving forest structural attributes that can be used for modelling, or is the focus of the study on deriving novel insights into bark beetle dynamics? The manuscript is very technical and presents a lot of technical details. The main conclusion also is that drones can be “a valuable tool for investigating multi-scalar phenomena...” (L. 533). I hence suspect that the main aim is presenting a novel methodological advancement, rather than generating exiting new insights into bark beetle dynamics. I encourage the authors to be crystal clear on the aim of their study.*

Our main goal was to assess how broad-scale environmental conditions interacted with local forest structure and composition to affect the probability of tree mortality during the cumulative (2012-2018) tree mortality event. The methodological advances in drone forest ecology were a necessary means by which we were able to meet this goal. A challenge in understanding bark beetle dynamics in forests (as you allude to) is a trade-off between the grain size and spatial extent of data that can be collected, and we feel that we have met that challenge in a novel way which has yielded new insights into the tree mortality patterns observed in the recent hot drought.

We have reframed the introduction and discussion to focus more on the ecology question being addressed and to de-emphasize the drone methodology (though the methods section remains drone-heavy by necessity).

R1.3: 2) *The authors state that using drones can enhance the set of predictors commonly used for predicting bark beetle outbreak dynamics by moving from stand-based to tree-based variables (L. 88). In their analysis, however, they only use very generic indicators (tree height, proportion of ponderosa pine and tree density), which have been investigated in previous studies already. They also aggregated tree-based measures to a common grid (level 4), which is somehow similar to doing analysis at the stand or plot scale (even though a more localized definition of stands). I thus have trouble seeing where the drone-based data offers advantage over more traditional approaches, e.g., using Lidar-based canopy height models or plot-based field measures of tree height, density, species, etc.*

The indicators that we chose to model can be considered generic, but they have clear relevance for bark beetle dynamics (as you allude to) at local scales. We see the strength of our study in its ability to capture both these fine scale measures simultaneously with the broad scale climatic water deficit gradient, and to do so with enough data at both scales to investigate their relative importance (and their interaction) in a rigorous way.

LiDAR measurements can generally capture forest structure well, but struggles with detecting individual dead trees or distinguishing between species (Jeronimo et al., 2019). Both of these challenges would be fatal flaws for our study, but can be overcome using our drone methodology using the much denser point clouds generated from structure from motion photogrammetry as well as passive reflectance data from a multispectral sensor. Finally, we roughly estimate that it would take 32 crews a full season of plot-based field sampling to acquire similar data to what we collected with 1-2 people over one summer.

Small, unhumanned aerial systems (sUAS) enable relatively fast and cheap remote imaging over hundreds of hectares of forest, which can be used to measure complex forest structure and composition at the individual tree scale with Structure from Motion (SfM) photogrammetry [@morris2017; @shiklomanov2019]. The ultra-high resolution of sUAS-derived measurements as well as the ability to incorporate vegetation reflectance can help overcome

challenges in species classification and dead tree detection inherent in other remote sensing methods, such as airborne LiDAR [@jeronimo2019].

R1.4: 3) *The questions asked in the introduction are good, but many of them could have been answered based on previous research results. The effect of proportion of host trees and tree density has been shown in many previous studies located in the US, Canada and Europe. Likewise, the positive link between drought and bark beetle activity is also well established in the scientific literature. Novel insights emerge from the interaction of drivers, but the general idea of cross-scale interactions has been established and tested in previous studies (i.e., Raffa et al. 2008, Seidl et al. 2016, Senf et al. 2017).*

We agree that some of the individual pieces of our work have well-founded explanations from smaller-scale work (e.g., the mechanisms underlying the effect of proportion of host trees on host tree mortality), or broader scale work (e.g., the influence of drought on bark beetle dynamics). We also agree that the notion of cross-scale interactions has been established in some previous work (we do cite Raffa et al., 2008, and Seidl et al., 2016 and we now cite Senf et al., 2017-- thanks for the recommendation!). However, as suggested by reviewer comment R2.1, our approach quantifies the cross-scale drivers of bark beetle-induced mortality using similarly-broad climate data as the studies you highlight, but much finer-scale forest structure data than these studies. We have reframed our introduction to focus much more heavily on refining our understanding on patterns of tree mortality observed during the California hot drought, and specifically how understanding the cross-scale interactions helps us better characterize mortality estimates from efforts that focus mostly on broader-scale environmental gradients or correlates of tree stress (e.g., local competitive environment).

R1.5: *Overall, my critique on very high level, and I want to emphasize that I really enjoyed reading the paper. I think this is an interesting and robust study that deserves publication. However, I think that the paper would be better suited for a more specialized journal, focusing either on the methodological advances or on the ecological insights generated.*

Thank you very much for the feedback! We still think Nature Communications is a great outlet for this work given the combination of ecological insights gained and methodological advances required to gain those insights. We have re-factored the framing of the paper in an attempt to highlight that it is the new ecology plus new methods in concert that make this work compelling to the audience of Nature Communications (especially in response to R1.2) and we hope you agree.

R1.6: L. 53: *Why are “primary” and “mass attack” in quotation marks?*

It was our intention to highlight for a more general ecology audience that these terms have specific meanings in the bark beetle/forest ecology literature, but we've removed them so as not to add unnecessary confusion.

R1.7: *Materials and Methods: To my knowledge, the Nature-family doesn't allow figures in the M&M section.*

We think the figures are particularly useful for understanding our methodological approach (as, of course, you point out in R1.8), but will defer to the editors if they need to be relocated to fit Nature-family style guidelines.

R1.8: *Figure 2: Great figure, it really helps understanding what you do!*

Thank you! We hope it helps ecologists new to UAV workflows get a quick sense of the mental model that underlies the data processing.

R1.9: *L. 313ff: What is the time frame of the analysis? Is the CWD data annual or are you using averages? If averages, what is the time frame the averages were calculated for? Is it 1981-2010? If so, how does this relate to the current drought? Some more information needed here.*

The data on forest structure were collected during a single field season in 2018 (all sites collected within a 3-month period) and thus represents a retrospective analysis of cumulative mortality throughout the hot drought. We have amended the text in the M&M subsection entitled “Aerial data collection and processing” to read:

All images were captured in 2018 during a 3-month period between early April and early July, and thus our work represents a postmortem investigation into the drivers of cumulative tree mortality.

The CWD data is a site-level (one value for each of the 32 sites) 30-year average from 1981 to 2010, which we note in the first sentence of the M&M subsection entitled “Environmental data”. The current drought was considered extreme across the extent of the CWD gradient. We make this more clear in the M&M subsection entitled “Environmental data”. The section now reads:

We used CWD (Stephenson 1998) from the 1981-2010 mean value of the basin characterization model (Flint et al. 2013) as an integrated measure of temperature and moisture conditions for each of the 32 sites. Higher values of CWD correspond to hotter, drier conditions and lower values correspond to cooler, wetter conditions. CWD has been shown to correlate well with broad patterns of tree mortality in the Sierra Nevada (Young et al. 2017) as well as bark beetle-induced tree mortality (Millar et al. 2012). The forests along the entire CWD gradient used in this study experienced exceptional hot drought between 2012 to 2015 with a severity of at least a 1,200-year event, and perhaps more severe than a 10,000-year event (Griffin and Anchukaitis 2014; Robeson 2015).

R1.10: *L. 337ff: Why do you assume that the zeros emerge from another process? Have you tested the model without zero-inflation term? Ecologically, it could be the same predictors that explain the absence of infestation (i.e., no host trees). Moreover, I didn't see where you report results for parameter p , so the proportion of plots experiencing no mortality. Maybe I missed it, but this would be interesting to see.*

We know that there is some misclassification error associated with the data that we are modeling, so we allow for an additional process (the zero-inflated part, which is intentionally decoupled from any ecological processes) to generate 0 values. For example, if there were actually dead trees in a cell, but we classified them all as alive, then we would have an erroneous 0 as a response variable for that cell which arose from a non-ecological process. We did test this model without the zero-inflated piece, and found much better success in the posterior predictive checks when including it (see note on posterior predictive checks in our response to R1.11).

The parameter 'p' does not represent the total proportion of plots with 0 mortality, but rather the probability of a plot having 0 dead ponderosa pine trees as a result of a separate, unmodeled process. A response of zero mortality can arise from the Binomially-distributed data model as well (with moments modeled as a linear combination of ecological processes). Because there is no process model underlying the 'p' parameter, we chose not to present it in the main text. However, we now present it in the supplemental information.

R1.11: L. 345: *I appreciate that you used Bayesian statistics and Stan to do so. That kind of models really show the power of Stan and Bayesian stats in general! However, I miss some crucial information on the MCMC sampling. Please note that you used MCMC samplers implemented in Stan and cite Carpenter et al. 2017. Second, you do not mention any priors, even though this is a crucial part of Bayesian analysis. Even if you use weakly informative priors as implemented in brms, report those in the paper as this is essential for reproducibility. Finally, I really appreciate you did posterior predictive checks, but I would love to see the graphs in the supplement.*

Re: software citations. Thanks for catching this. We are very keen to cite software and statistical methods that we used. After describing the model in Equation 1, we now say:

We fit this model using the `brms` package (Bürkner 2017) which implements the No U-Turn Sampler extension to the Hamiltonian Monte Carlo algorithm (Hoffman & Gelman 2014) in the Stan programming language (Carpenter et al. 2017).

Re: priors. We agree these need to be specified. We did indeed use weakly-regularizing default priors from the `brms` package and now say:

We centered and scaled all predictor values, and used weakly-regularizing default priors from the `brms` package (Bürkner 2017).

Re: posterior predictive checks. We also now include the plot of our posterior predictive check in the supplemental information.

R1.12: L. 386: *Throughout the discussion, you always start a paragraph by repeating the results. This is very repetitive, and I suggest using each paragraph's topic sentence for highlighting the main implications of the results, rather than repeating what was reported in the results section already.*

Thanks for the suggestion, we have removed these topic sentences from the Discussion (in favor of reporting the more detailed information about effect sizes contained within them to the results). The discussion section paragraphs now begin with the relevance of each finding, rather than the finding itself.

RESPONSES TO REVIEWER #2

R2.1: *"Cross-scale interaction ... " addresses an important concern, increased losses of forest resources to native insects whose populations are exacerbated by a changing environment. The study focuses on mortality of ponderosa pine to western pine beetle in California USA, but has implications to bark beetles worldwide.*

A major strength of the paper is its emphasis on cross-scale interactions, which are highly difficult to partition despite their widely recognized importance. A strength of the experimental

approach is the use of drones to provide spatially explicit data. This approach also brings a significant weakness, namely that all dead trees were assumed to be ponderosa pine, and all of these were assumed to have been killed by western pine beetle. However, the authors recognize this limitation and allot it ample discussion. If it could also be included in the Abstract that would be better yet. The paper is well written. The statistical analyses are appropriate. Despite this paper's strengths it requires some improvement before it can be acceptable for publication in Nature Communications. I identify several issues below, each of which I feel can be achieved in a revision.

Thank you very much for this feedback. We agree that the focus on cross-scale interactions is a key piece to our manuscript, and think a major contribution of our work is in overcoming the challenge you note-- that they are generally difficult to partition.

We added our classification caveat to our abstract, which now reads:

We use drone surveys over 32 distinct sites along a 350-km latitudinal and 1000-m elevational gradient in western slope Sierra Nevada ponderosa pine/mixed-conifer forests and structure from motion (SfM) processing to segment and classify more than 450,000 trees over 9 km² of forest with WPB-induced tree mortality. We validated the segmentation and classification with data from 160 coincident field plots (each 0.041 ha in area) throughout the 32 sites, assuming that dead trees were all ponderosa pine killed by WPB.

R2.2: 1. *The title refers to 'Climate', but the paper is about Climatic Water Deficit. Climate is a much more expansive term, including temperature (which also affects the beetles directly not just via the host), wind, etc., each of which influences specific features of bark beetle population dynamics. So 'Climate' should be replaced with either 'Climatic Water Deficit' or 'Drought' in the title. This is a simple, but important fix.*

We are on board with using 'climatic water deficit' in the title, but opted for 'climate' given the more general audience of Nature Communications as well as the Nature-family limitations on total title length. We will leave it as 'climate' for now and hope we can convince you that the more general term will still suffice given that climatic water deficit integrates two of the primary climate variables important for plants-- temperature and precipitation-- especially in the semi-arid forests of the western Sierra Nevada. 'Drought' doesn't capture the key message of the paper, because the drought was a background phenomenon across the whole study area, while CWD represents the historic water availability conditions at each site.

R2.3: 2. *The second problem is more systemic, and will require some substantial reframing. Specifically, the literature already provides mechanistic explanations for some of the trends reported here, so the appearance of these trends is less surprising than suggested, although I agree they go against conventional dogma. These mechanistic studies should be better integrated into the story.*

Thank you for the suggestion. We added new text early in the introduction in order to appropriately frame the paper as an investigation into incompletely-explained tree mortality patterns during California's hot drought. Specifically, we say:

Tree mortality exhibited a strong latitudinal and elevational gradient (Asner et al. 2016, Young et al. 2017) that can only be partially explained by coarse-scale measures of

environmental conditions (i.e., historic climatic water deficit; CWD) and current forest structure (i.e., current regional basal area) (Young et al. 2017). Progressive loss of canopy water content offers additional insight into tree vulnerability to mortality, but cannot ultimately resolve which trees die in forests with bark beetles as a key mortality agent (Brodrick and Asner 2017). Bark beetles respond to local forest characteristics in positive feedbacks that non-linearly alter tree mortality dynamics against a background of environmental conditions that stress trees (Raffa et al. 2008, Boone et al. 2011). Thus, an explicit consideration of local forest structure and composition (Stephenson et al. 2019, Fettig et al. 2019) as well as its cross-scale interaction with regional climate conditions (Senf et al. 2017) can refine our understanding of tree mortality patterns from California's recent hot drought. The challenge of simultaneously measuring the effects of both local-scale forest features (such as structure and composition) and broad-scale environmental conditions (such as climatic water deficit; CWD) on forest insect disturbance leaves their interaction effect relatively underexplored (Seidl et al. 2016, Senf et al. 2017, Stephenson et al. 2019, Fettig et al. 2019).

To your point about mechanistic explanations already being known for bark beetle dynamics, we agree that some of our findings are not surprising. However, we assessed how broad-scale environmental conditions interacted with local forest structure (and composition) to affect the probability of tree mortality during the cumulative (2012-2018) tree mortality event. Our work was executed in an area of complex topography with a Mediterranean climate during what is now considered a 1,200-year drought event. While the literature provides some mechanistic explanations for some of the trends observed, this literature comes from other bark beetle-host systems involving different ecologies. To that end, we have integrated the references that you mention, and now try to strike a balance between presenting western pine beetle-specific ecology (and gaps in knowledge) with borrowing insights from other bark beetle-host systems (i.e., primarily mountain pine beetle in lodgepole pine in the Intermountain West) on which many of these mechanics manuscripts are based. The revised text points out that the inciting factors for each system are quite different. For mountain pine beetle, the inciting factor is typically temperature (both in terms of its influence on overwintering success and adaptive seasonality) and drought generally has much less of an influence (especially in lodgepole pine). For western pine beetle, the inciting factor is drought stress (d X t X competition) as manifested through changes in host tree physiology and thus susceptibility. We hope we've balanced this well, and in doing so, have highlighted why the results we present are indeed valuable and unique contributions to the understanding of WPB in California.

R2.4: *a. For example, the contrasting relationships that host susceptibility and host brood quality have with tree size, and how beetle choice transitions with its population numbers, are mechanistically delineated in Boone et al. 2011. This is also described much more mechanistically and at finer-scale resolution in Raffa et al. 2008, 2015 than depicted here.*

We agree that this is a good section in which to highlight the work of Boone et al., 2011 and we now cite it there.

R2.5: *b. Likewise, the positive effect of host density on beetle attack success can at least partly be attributed to their switching behavior whereby aggregated beetles move to adjacent trees. This is described by Geizler & Gara 1978, Mitchell & Preisler 1991 and Preisler 1993, which should be cited. Also empirical evidence for tree density decreasing defense capacity is described in fir by Raffa & Berryman 1982.*

We agree, and have now cited Geizler & Gara 1978, Mitchell & Preisler 1991, and Preisler 1993. As far as we can tell, Raffa & Berryman 1982 presents evidence for defense capacity in suppressed vs. non-suppressed trees, and shows a relationship between the distance to the nearest neighbor and defence capacity only for the suppressed trees (but not intermediate or dominant trees). We have decided to leave out this reference in favor of the other, more relevant work.

R2.6: c. *Important mid-scale data on relationships between moisture availability and bark beetle attack are in Kaiser et al. 2012. These should be cited.*

This is a great paper and we now cite it.

R2.7: 1. *Line 56: 'This Allee effect creates a strong coupling between beetle selection behavior of host trees and host tree susceptibility to colonization' Perhaps it would be better to state 'This Allee effect creates a strong coupling among beetle selection behavior of host trees, host tree susceptibility to colonization, and beetle stand-level density'. Include Wallin & Raffa 2004, as that provides empirical data for how bark beetles change host selection behavior with population density.*

We now include Wallin & Raffa 2004 in this sentence, as it is a great example of the consequences of the coupling between beetle choice and host susceptibility. We would argue that the Allee effect, being a form of density dependence by definition, already implies a role of beetle stand-level density so we have kept the list of things that are coupled to just the beetle decision making and the host tree susceptibility.

R2.8: 2. *Line 57: Resin physically expels, or delays, beetles but it also contains chemicals that are toxic to the beetles and their fungi.*

Good point. This line now reads:

A key defense mechanism of conifers to bark beetle attack is to flood beetle bore holes with resin, which physically expels colonizing beetles, can be toxic to the colonizers and their fungi, and may interrupt beetle communication [@franceschi2005; @raffa2015].

R2.9: 3. *Line 59, 66, 77, 475: Some examples where Boone et al. 2011 seems particularly relevant.*

We mostly agree. These lines now read:

L59: Under normal conditions, weakened trees with compromised defenses are the most susceptible to colonization and will be the main targets of primary bark beetles like WPB (Bentz et al., 2010; Boone et al., 2011; Raffa et al., 2015).

L66: As the local population density of beetles increases due to successful reproduction within spatially-aggregated weakened trees, as might occur during drought, mass attacks grow in size and become capable of overwhelming formidable tree defenses such that even healthy trees may be susceptible to colonization and mortality (Bentz et al., 2010; Boone et al., 2011; Raffa et al., 2015).

L77: Tree size is another aspect of forest structure that affects bark beetle host selection behavior with smaller trees tending to have lower capacity for resisting attack, and larger trees being more desirable targets on account of their thicker phloem providing greater nutritional content (Chubaty et al., 2009; Boone et al., 2011; Graf et al., 2012).

L475 The references in this line are specifically related to other studies examining Sierra Nevada tree mortality during the 2012 to 2015 hot drought, so we chose not to cite Boone et al., 2011 here.

R2.10: *Line 88-91: Kaiser et al. 2012 seems particularly relevant here.*

We agree. This line now reads:

Stand-scale measures of forest structure and composition thus paint a fundamentally limited picture of the mechanisms by which these forest characteristics affect bark beetle disturbance, but finer-grain information explicitly recognizing tree size, tree species, and local tree density should more appropriately capture the ecological processes underlying insect-induced tree mortality (Kaiser et al., 2013).

R2.11: 4. *Line 146-7. It would be more accurate to state nonhost volatiles inhibited attraction of WPB to its aggregation pheromone, as is stated in lines 408-9.*

We've taken your suggestion for clarification:

Volatiles from several nonhosts sympatric with ponderosa pine have been demonstrated to inhibit attraction of WPB to its aggregation pheromones (Fettig et al., 2015; Shepherd et al., 2007).

R2.12: 5. *Line 149-50: Should something be mentioned here about other competing bark beetle species besides mountain pine beetle?*

We restrict our mention to just the MPB as the only other primary bark beetle that regularly can be found in ponderosa pine in this range. Fettig et al. (2019) examined 1,695 ponderosa pine trees that died across their network of plots and only attributed mortality to mountain pine beetle on 14 occasions, which they found to be a surprisingly low number. They did attribute ponderosa pine mortality to engraver beetle (*Ips* spp.) on 36 occasions, but these species tend to not be considered as aggressive as WPB or MPB in this range, and tree mortality attributed to *Ips* spp. was concentrated in small-diameter trees. We did modify our text to make this more clear:

In California, WPB generally has 2-3 generations in a single year and can often out-compete other primary bark beetles such as the mountain pine beetle in ponderosa pines, especially in larger trees (Miller and Keen, 1960).

R2.13: 6. *Fig. 2: I really like the extensive figure caption you provide here!*

Thanks! We hope it provides a mental model for folks new to thinking about how drone-derived data can augment field sampling in forest ecology. See also our responses to comment R1.7 and R1.8.

R2.14: 7. *Lines 420-425. The positive relationships you describe between tree growth and tree resistance seem important and perhaps merit stronger emphasis.*

We have tried to make this paragraph more clear in order to emphasize the point that density is itself a consequence of microclimate and thus denser patches might indicate more favorable site conditions (and be home to more resistant trees). Per Reviewer 3's suggestion, we have also reworked our model to include basal area and we hope that the discussion of that effect helps to contextualize the points made in this paragraph.

RESPONSES TO REVIEWER #3

R3.1: *The aims of the study were to develop methods for estimating tree mortality using drone photographs across a broad gradient of ponderosa and mixed conifer forests in the Sierra Nevada Range and determine if local tree height, number of trees and estimated long-term site water availability could be related to tree mortality. The methods developed showed that plot characteristics and tree mortality could be modestly estimated using the drone imagery and post-processing steps, compared to the 110 ~400 m² ground-measured plots visible in the drone imagery. Correlations with plot characteristics and drone imagery ranged from 0.16 for 25th percentile height to 0.67 for total tree height. No comparison for drone-based estimates of mortality to the ground-measured plots was given, though the study claimed 98% success in estimating mortality in a data set withheld from developing the models.*

We compared the drone-based estimates of mortality to the ground-based measures in a supplemental table, and now highlight that more prominently in the Results section under the "Site summary based on best tree detection algorithm and classification" subsection where we say:

See Supplemental Information for site summaries and comparisons to site-level mortality measured from field data.

R3.2: *Using the drone data set of plot characteristics, and an areal description of estimated long-term site water availability, models estimated that the proportion of ponderosa pine host trees had the largest effect (positive) on mortality, followed by the intercept and ponderosa mean height (both negative effects), and the combination of ponderosa mean height and estimated long-term site water availability (positive effect). Tree density and the combination of tree density and estimated long-term site water availability had small negative effects. Overall, the study suggests that drone imagery can be helpful in identifying some of the stand characteristics that relate to local variability in bark-beetle mortality, but the results are puzzling compared to earlier work (Waring and Pitman 1985), and there are some fairly major issues that need to be addressed.*

We generally agree with this summary, and have taken steps to reduce the focus on the methodology development in favor of the ecological insights (see also our response to comment R3.3).

We don't find our results to be in opposition to those of Waring and Pitman (1985), though we do now cite this work. Waring and Pitman (1985) used manipulations of forest structure in an experimental framework to attribute treatment effects to mountain pine beetle-induced lodgepole pine mortality. Waring and Pitman (1985) showed that a reduction in local tree density clearly reduced the probability of lodgepole pine mortality in a mountain pine beetle outbreak. We did find that more ponderosa pine trees (the WPB host) increased the mortality

rate of ponderosa pine, but that the existing, mostly natural gradient of overall forest density (including all species) negatively correlated with ponderosa mortality rates. This finding (greater overall density reduces ponderosa pine mortality rates) is corroborated by Restaino et al., 2019 (in the part of their study that looks at natural gradients of forest density) and Fettig et al., 2019. We attribute this finding the same way that Restaino et al., 2019 do: overall local density likely being a good indicator of favorable microsite conditions in these fire suppressed forests, and the trees in those favorable microsites being more resistant to attack.

Notably, the strength and direction of the forest density effect in Waring and Pitman (1985) is more similar to that of Restaino et al., (2019) (in the part of their study that examines the effects of recent forest thinning treatments on bark beetle-induced mortality).

This section now reads:

The negative relationship between overall tree density and the probability of ponderosa pine mortality corroborates findings of coincident ground plots [@fettig2019, in their analysis using proportion of trees killed as a response] and other work during the same hot drought [@restaino2019].

The forest structure (in the absence of management) is itself a product of climate and, with increasing importance at finer spatial scales, topographic conditions [@fricker2019].

Thus, the denser forest patches in our study may indicate greater local water availability, more favorable conditions for tree growth and survivorship, and increased resistance to beetle-induced mortality [@ma2010; @fricker2019; @restaino2019].

R3.3: 1. Tree density (used in this paper) is a very poor surrogate for the influence of surrounding trees on the resources of a target individual. Plot basal area is a much better surrogate for the overall influence of surrounding trees on the trees in a plot, because basal area is closely related to both leaf area and biomass—which are both closely related to resource use. The number of trees can be related to resource impact, but only if there are an extraordinary number of very small trees or if there are a large number of large trees. Perhaps this is why the results for 'density' and tree height run counter to earlier studies (Waring and Pitman 1985). Basal area needs to be used!

Thanks for the suggestion. We have reworked our model to include basal area. We still include overall density, given its ability to influence WPB dynamics via mechanisms other than those mediated by local tree competition. Our negative effect of overall tree density remains, but does not surprise us considering the explanation we give in the section referenced by our response to R2.14. In short, denser patches are likely areas with more favorable microclimate in these fire-suppressed forests (Ma et al., 2010; Fricker et al., 2019) and are more likely to have more resistant trees. This is a similar explanation as offered by other studies showing a similar response to tree density in similar forests during the same hot drought (Restaino et al., 2019). Further, it does not surprise us that our results differ from those of Waring and Pitman, 1985, who manipulated tree density experimentally rather than using natural gradients to make inferences.

We offer several possible explanations for why we find a negative relationship between ponderosa pine size and mortality probability, and added another one based on work that Reviewer 1 suggested we cite. That section now reads:

The negative main effect of host tree mean size was surprising, and appears to contradict long-standing wisdom on the dynamics of WPB in the Sierra Nevada. WPB exhibit a preference for trees 50.8 to 76.2 cm DBH (Person 1928, 1931), and a positive relationship

between host tree size and levels of tree mortality attributed to WPB was reported by Fettig et al. (2019) in the coincident field plots as well as in other recent studies (Restaino et al. 2019, Stephenson et al. 2019, Pile et al. 2019). Indeed, Fettig et al. (2019) attributed no mortality to WPB in ponderosa pine trees <10.0 cm DBH and found no tree size/mortality relationship for incense cedar or white fir in the coincident field plots. These species represent 22.3% of the total tree mortality observed in their study, yet in our study all dead trees were classified as ponderosa pine (see Methods) which could dampen the positive effect of tree size on tree mortality. Larger trees are more nutritious and are therefore ideal targets if local bark beetle density is high enough to successfully initiate mass attack and overwhelm tree defenses, as can occur when many trees are under severe water stress (Bentz et al. 2010, Boone et al. 2011, Kolb et al. 2016). In the recent hot drought, we expected that most trees would be under severe water stress, setting the stage for increasing beetle density, successful mass attacks, and targeting of larger trees. A possible explanation for our finding counter to this expectation is that our observations represent the cumulative mortality of trees during a multi-year drought event and its aftermath. Lower host tree mean size led to a greater probability of host mortality earlier in this drought (Pile et al. 2019, Stovall et al. 2019) and that signal might have persisted even as mortality continued to accumulate driven by other factors. Another explanation may be that our extensive sampling design better captured the contagious process by which bark beetles colonize smaller, suboptimal trees in the vicinity of the larger, more desirable trees that are the focus of initial attack (e.g., Klein et al. 1978). If larger, desirable trees tend to be associated with a greater local density of smaller trees that are also colonized in this contagious process, then we might observe a negative relationship between tree size and ponderosa mortality rates. Finally, tree growth rates may be a better predictor of susceptibility to WPB colonization than tree size per se, with slower-growing trees being most vulnerable (Miller and Keen 1960). While slow-growing trees are often also the largest trees, this may not be the case for our study sites especially given the legacy of fire suppression in the Sierra Nevada and its effect of perturbing forest structure far outside its natural range of variation (Safford and Stevens 2017).

R3.4: 2. Proportion of tree mortality relative to total trees in a plot is also a very poor surrogate for the effects of that mortality on the ecosystem. It is not the number of trees killed or the relative number of trees that are killed that is important to ecosystem effects such as water use, streamflow, or carbon storage, it is the proportion of basal area that is lost. The use of proportional number of trees killed instead of the proportion of basal area killed may also have contributed to the odd results (tree density and mean height having a negative effect on mortality fraction), and compared to earlier studies. Again, basal area would be much more biologically meaningful than tree numbers.

Our primary goal was to understand the dynamics of the western pine beetle, rather than to characterize the effects of tree mortality on the ecosystem. We agree that the probability of ponderosa pine mortality wouldn't be the best choice for a response variable if the latter were our ultimate goal. We take your point and add to our "Limitations" section in our discussion:

Finally, we note our study is constrained by using the probability of ponderosa mortality as our key response variable. This measure is well-suited to understanding the dynamics between WPB colonization behavior and host tree susceptibility, but may not capture impacts on the forest ecosystem and its services as well as a measure of biomass reduction such as tree basal area.

R3.5: 3. Data for the model estimates of mortality needs to be presented for the ground-measured plots, and the method for estimating mortality for the models needs to be presented.

We present information on the classifier for estimating individual tree mortality (i.e., the method for estimating mortality for the models) in the methods section:

For each crown polygon, we calculated the mean value of the extracted Level 2 and Level 3a pixels and used them as ten independent variables in a five-fold cross validated boosted logistic regression model to predict whether the hand classified trees were alive or dead.

We estimate mortality at each site using this model, and present the overall mortality estimate along with the field-measured mortality in a supplemental table (see also our response to R3.1).

R3.6: *Finally, 'CWD' is a poor choice of acronym for a study on forests, where it usually means 'Coarse Woody Debris' or 'Chronic Wasting Disease'.*

Sadly, CWD is what we are stuck with as “climatic water deficit” is a relatively common term in a more general ecology sense as is the use of its “CWD” acronym. We hope the general audience of Nature Communications can look past this!

Literature to cite (per reviewer recommendations)

Boone et al. 2011. Efficacy of tree defense physiology varies with bark beetle population density: A basis for positive feedback in eruptive species. *Canadian Journal of Forest Research*, 2011, 41(6): 1174-1188, <https://doi.org/10.1139/x11-041>

Carpenter, B. et al. 2017. Stan: A Probabilistic Programming Language. *Journal of Statistical Software* 76, 1–32. <https://doi.org/10.18637/jss.v076.i01>

Geiszler, D. R. and R. I. Gara. 1978. Mountain pine beetle attack dynamics in lodgepole pine. In *Theory and Practice of Mountain Pine, Beetle Management in Lodgepole Pine Forests: Symp. Proc.* A. A. Berryman, G. D. Amman and R. W. Stark (Eds). Pullman, WA: Washington State Univ.

Kaiser et al. 2013. Ecohydrology of an outbreak: Mountain pine beetle impact trees in drier landscape positions first. <https://doi.org/10.1002/eco.1286>

Mitchell & Preisler 1991. Analysis of spatial patterns of lodgepole pine attacked by outbreak populations of the mountain pine beetle. <https://doi.org/10.1093/forestscience/37.5.1390>

Preisler 1993. Modelling spatial patterns of trees attacked by bark beetles. *Journal of the Royal Statistical Society. Series C (Applied Statistics)*. <https://doi.org/10.2307/2986328>

Raffa & Berryman 1982. Accumulation of monoterpenes and associated volatiles following inoculation of grand fir with a fungus transmitted by the fir engraver, *Scolytus ventralis* (Coleoptera: Scolytidae). <https://doi.org/10.4039/Ent114797-9>

Raffa, K. F. et al. 2008. Cross-scale Drivers of Natural Disturbances Prone to Anthropogenic Amplification: The Dynamics of Bark Beetle Eruptions. *BioScience* 58, 501–517. <https://doi.org/10.1641/B580607>

Raffa et al., 2015 Chapter 1 - Natural history and ecology of bark beetles. In Bark Beetles: Biology and ecology of native and invasive species. <https://doi.org/10.1016/B978-0-12-417156-5.00001-0>

Seidl, R. et al. Small beetle, large-scale drivers: how regional and landscape factors affect outbreaks of the European spruce bark beetle. *J Appl Ecol* 53, 530–540 (2016). <https://doi.org/10.1111/1365-2664.12540>

Senf, C., Campbell, E. M., Pflugmacher, D., Wulder, M. A. & Hostert, P. A multi-scale analysis of western spruce budworm outbreak dynamics. *Landscape Ecology* 32, 501–514 (2017). <https://doi.org/10.1007/s10980-016-0460-0>

Wallin & Raffa 2004. Feedback between individual host selection behavior and population dynamics in an eruptive herbivore. <https://doi.org/10.1890/02-4004>

Waring RH, Pitman GB (1985) Modifying lodgepole pine stands to change susceptibility to mountain pine beetle attack. *Ecology* 66:889-897. <https://doi.org/10.2307/1940551>

Reviewer comments, second round –

Reviewer #1 (Remarks to the Author):

Thank you very much for answering and addressing all of my comment. You have done a great job in revising the manuscript.

Reviewer #2 (Remarks to the Author):

This revision shows significant improvement over the initial version. And its major strength, the use of a novel experimental approach to help address the highly difficult but important goal of partitioning drivers across ecological scales, continues to give this paper high potential. Also, the organized responses to queries by myself and the other reviewers are very helpful for evaluation. In addition to the improvements, however, some important concerns remain and these need to be addressed before this ms is suitable for Nature Communications.

1. The term 'climate' in the title overstates the breadth of the study. I empathize with how you want your title to appeal 'to the more general audience of Nature Communications', but the paper's actual breadth is fine in my view and in the view of one of the other reviewers. Also, the concern about a more accurate title exceeding journal length limits isn't warranted because changing 'climate' to 'climatic water deficit' only raises the word count to 15, the journal limit. The title change is more than just semantics because three separate components of climate, specifically high temperatures, drought, and windstorms, have already been shown to influence multiple bark beetle systems and to do so by distinct (sometimes interacting) processes. So synonymizing just one of those components with the overarching term 'climate' detracts from that understanding.
2. The second major problem, inadequately connecting the novelty of this draft's findings to previous mechanistic understanding, has been partially, but not entirely, addressed. This point was also identified as a major concern by Reviewer 1. The revision now adds some pertinent literature, but at times these references seem more appended onto others than fully integrated into the framework. I give three examples below:
 - a. In trying to address reviewers' request to better portray mechanistic understanding the authors inadvertently create a new problem by depicting a dichotomy between western pine beetle and mountain pine beetle (L73-75). In addition to undermining the breadth they're trying to convey for Nature Communications (i.e., the paper now veers to being just about western pine beetle in California), this binary distinction is not tenable. Many analyses clearly establish both high temperatures (Powell & Bentz 2009, Sambaraju et al. 2019) and drought (Kaiser et al. 2013, Raffa et al. 2008, Sambaraju et al. 2019) as important drivers of mountain pine beetle outbreaks. Moreover, the roles of multiple drivers applies broadly across most primary bark beetle species, e.g., North American spruce beetle being driven by both temperature (Werner & Holsten 1985, Raffa et al. 2008, DeRose et al. 2012) and drought (Hart et al. 2014, 2017), and European spruce beetle being driven by temperature (Marini et al. 2017), drought (Netherer et al. 2019), and windstorms (Marini et al. 2017, Netherer et al. 2019). So it is best to delete the portion in parentheses and perhaps include pertinent references from other systems.
 - b. At times this ms conflates stand-level (forest structure) and tree-level (defense) processes. An example is L104: "Stand-scale measures of forest structure and composition thus paint a fundamentally limited picture of the mechanisms by which these forest characteristics affect bark beetle disturbance, but finer-grain information explicitly recognizing tree species, size, and local density should better capture the ecological processes underlying insect-induced tree mortality (Geiszler & Gara 1978, Mitchell & Preisler 1991, Preisler 1993, Kaiser et al. 2013)." The references I suggested that are now cited pertain to stand-level processes, which is good. But the omitted reference (Raffa & Berryman 1982) provides tree-level (i.e., finer-grain) mechanisms to help fill that 'limited picture'. That paper's finding that suppressed trees had weaker biochemical defenses than dominant/codominant trees, and that within suppressed trees crowding reduces defense, seems to be provide 'finer-grain information' consistent with the current ms' findings about relationships between tree size and beetle-caused mortality. Also, an important paper that has

helped clarify stand-level processes is Faccoli & Bernardinelli 2014.

c. The discussion places strong emphasis on the negative main effect of host tree size, and how the relationship of host tree size to tree mortality varies with environmental stress. This is indeed new and important information for western pine beetle. However, restricting this discussion to western pine beetle makes the ramifications seem needlessly narrow for Nature Communications and detached from related systems. It misses an opportunity to update long-held views (conventional wisdom) of host size more generally by not pointing to the consistency of the current results with results on mountain pine beetle when analyses of host size effects are linked to tree defense physiology (Boone et al. 2011).

3. Line 495: It is probably better to say 'conventional wisdom' than 'wisdom'.

4. Line 171-172: Yes, western pine beetle outcompetes mountain pine beetle for live ponderosa pines, but because this paragraph describes population dynamics rather than tree mortality my comment about other competitors was meant to include beetle development success following tree death. Secondary bark beetles such as *Ips* and *D. valens*, wood borers, and antagonistic fungi can reduce primary beetle reproduction through competition, even despite some spatial and temporal separation.

5. Line 565: You should clarify that lodgepole pine monocultures are from natural regeneration, as some readers might envision uniform plantings as with European spruce and loblolly pine. Also, it might protect you from criticism to say 'relative' monocultures or apply some other modifier.

6. Line 472: Two more appropriate references than Raffa & Berryman 1987 about thinning are Fettig 2007 and Fettig et al. 2014, so replace the former with one or both of the latter two.

Nice work.

Reviewer #3 (Remarks to the Author):

I reviewed the original submission of this manuscript, and appreciate the careful attention to all of the reviewer comments, including mine, in the revision. The methods are more clearly presented, and the method and results are very helpful and the manuscript will be an important contribution. My only suggestion is to reiterate my plea for presenting the mortality results as percent of 'plot' basal area in addition to the percent of numbers of trees:

From my original review: "2. Proportion of tree mortality relative to total trees in a plot is also a very poor surrogate for the effects of that mortality on the ecosystem. It is not the number of trees killed or the relative number of trees that are killed that is important to ecosystem effects such as water use, streamflow, or carbon storage, it is the proportion of basal area that is lost. The use of proportional number of trees killed instead of the proportion of basal area killed may also have contributed to the odd results (tree density and mean height having a negative effect on mortality fraction), and compared to earlier studies. Again, basal area would be much more biologically meaningful than tree numbers."

I get that the aim was focused on individual trees, but trees come in widely different sizes, such that the effects of the number or percent of trees killed in a stand will vary widely with the size of those trees. Basal area takes care of this issue and would be more useful to forest managers and forest ecologists. Since the authors included basal area as an estimate of competition, I believe that the authors have the data to estimate this metric and provide more useful information than numbers of trees or percent of total trees. Managers would be less concerned if the mortality were focused in just small trees and very concerned if it focused in just large trees. Likely it is both, but perhaps with a different pattern than for tree numbers. Doing basal area mortality might be analogous to identifying age and race for human Covid-19 mortality--people are roughly the same size, but age and race give the public health folks for information for management.

Mike Ryan

NCOMMS-20-01457B response to reviews

Dear Reviewers,

Thank you very much for your feedback.

Below, we address your comments point-by-point. As before, the exact text from your comments are copied in this document and are displayed in black italic font and have each been assigned a number for reference (SE.1 for the first point by the senior editor, R1.1 for the first point by reviewer one, R2.1 for the first point by reviewer 2, etc.). Our responses to these individual points are written in **blue text**, and any pertinent direct quotes from our revised manuscript that reflect changes addressing your comments are written in **green text**.

We also note that we implemented an additional calibration of our drone-measured tree heights using the field-measured tree heights, after noticing a systematic underestimate of the drone-estimated heights of dead trees. This had the effect of strengthening the positive main effect of CWD on ponderosa mortality probability in our model, and reversing the negative main effect of tree height. That is, we previously reported that greater tree height reduced the probability of ponderosa mortality. We now report, using the calibrated heights, that greater tree height increases the probability of ponderosa mortality. We intentionally used a conservative calibration, so our now-reported result of a positive effect of host tree height is likely a lower bounds on the true magnitude of the effect (and we can be additionally confident that the effect is not negative). We have more confidence in the calibrated tree heights, and these results now better align with known life history of the western pine beetle. The bias likely arises as narrow, needleless treetops of the standing dead trees can be missed by the Structure from Motion processing, effectively reducing their reported height. We detail this calibration in the methods, discussion, and supplemental materials (including a note on how the calibration affected our modeling results). The key result of our paper-- the cross-scale interaction between host tree size and climatic water deficit, remains strong.

Thank you again for your continued efforts to improve this manuscript.

Best,
The Authors

Reviewer #1 (Remarks to the Author):

R1.1: *Thank you very much for answering and addressing all of my comment. You have done a great job in revising the manuscript.*

Great! Thank you very much for your comments, as we think addressing them greatly improved the manuscript.

Reviewer #2 (Remarks to the Author):

R2.1: *This revision shows significant improvement over the initial version. And its major strength, the use of a novel experimental approach to help address the highly difficult but important goal of partitioning drivers across ecological scales, continues to give this paper high potential. Also, the organized responses to queries by myself and the other reviewers are very helpful for evaluation. In addition to the improvements, however, some important concerns remain and these need to be addressed before this ms is suitable for Nature Communications.*

Thank you for these further comments. We are also glad the format of our responses helps to facilitate dialogue during the review process. We have integrated the comments from you and Reviewer #3 into our manuscript.

R2.2: *1. The term 'climate' in the title overstates the breadth of the study. I empathize with how you want your title to appeal 'to the more general audience of Nature Communications', but the paper's actual breadth is fine in my view and in the view of one of the other reviewers. Also, the concern about a more accurate title exceeding journal length limits isn't warranted because changing 'climate' to 'climatic water deficit' only raises the word count to 15, the journal limit. The title change is more than just semantics because three separate components of climate, specifically high temperatures, drought, and windstorms, have already been shown to influence multiple bark beetle systems and to do so by distinct (sometimes interacting) processes. So synonymizing just one of those components with the overarching term 'climate' detracts from that understanding.*

We now better understand your perspective with respect to more precisely titling the manuscript and have changed the title to your suggestion. We were (wrongly) operating under the impression that there was a 100-character limit to the titles for *Nature Communications* (rather than, as you point out, a 15-word limit) and so we are happy to change our title to use "climatic water deficit" in place of "climate". Thanks for your reassurance that this term will still be accessible to the journal's audience!

R2.3: *2. The second major problem, inadequately connecting the novelty of this draft's findings to previous mechanistic understanding, has been partially, but not entirely, addressed. This point was also identified as a major concern by Reviewer 1. The revision now adds some pertinent*

literature, but at times these references seem more appended onto others than fully integrated into the framework. I give three examples below:

We have made some major changes to address this critique that we think better tie our findings to the broader literature of primary bark beetles (and not just the western pine beetle). We detail some of these changes for each of the examples that you give below.

R2.4: a. *In trying to address reviewers' request to better portray mechanistic understanding the authors inadvertently create a new problem by depicting a dichotomy between western pine beetle and mountain pine beetle (L73-75). In addition to undermining the breadth they're trying to convey for Nature Communications (i.e., the paper now veers to being just about western pine beetle in California), this binary distinction is not tenable. Many analyses clearly establish both high temperatures (Powell & Bentz 2009, Sambaraju et al. 2019) and drought (Kaiser et al. 2013, Raffa et al. 2008, Sambaraju et al. 2019) as important drivers of mountain pine beetle outbreaks. Moreover, the roles of multiple drivers applies broadly across most primary bark beetle species, e.g., North American spruce beetle being driven by both temperature (Werner & Holsten 1985, Raffa et al. 2008, DeRose et al. 2012) and drought (Hart et al. 2014, 2017), and European spruce beetle being driven by temperature (Marini et al. 2017), drought (Netherer et al. 2019), and windstorms (Marini et al. 2017, Netherer et al. 2019). So it is best to delete the portion in parentheses and perhaps include pertinent references from other systems.*

Thanks for this other context. We have deleted the parenthetical, de-emphasized the dichotomy between WPB and other systems, and added more of the work you reference to flesh out the introduction to primary bark beetle dynamics in a more general way. The particularly relevant paragraph now reads:

The ponderosa pine/mixed-conifer forests in California's Sierra Nevada region are characterized by regular bark beetle disturbances, primarily by the influence of western pine beetle (*Dendroctonus brevicomis*; WPB) on its host ponderosa pine (*Pinus ponderosa*) (Fettig 2016). WPB is a primary bark beetle—its reproductive success is contingent upon host tree mortality, which itself requires enough beetles to mass attack the host tree and overwhelm its defenses (Raffa and Berryman 1983). This Allee effect creates a strong coupling between beetle selection behavior of host trees and host tree susceptibility to colonization (Wallin and Raffa 2004, Raffa and Berryman 1983, Logan et al. 1998). A key defense mechanism of conifers to bark beetle attack is to flood beetle bore holes with resin, which physically expels colonizing beetles, can be toxic to the colonizers and their fungi, and may interrupt beetle communication (Franceschi et al. 2005, Raffa et al. 2015). Under normal conditions, weakened trees with compromised defenses are the most susceptible to colonization and will be the main targets of primary bark beetles like WPB (Raffa et al. 2015, Bentz et al. 2010, Boone et al. 2011). Under severe water stress

however, many trees no longer have the resources available to mount a defense (Kolb et al. 2016, Boone et al. 2011). Drought (Hart et al. 2017, Netherer et al. 2019, Raffa et al. 2008, DeRose and Long 2012), especially when paired with high temperatures (Marini et al. 2017, Bentz et al. 2010, Kaiser et al. 2013, Sambaraju et al. 2019), can trigger increased bark beetle-induced tree mortality as average tree vigor declines. As the local population density of beetles increases due to successful reproduction within spatially-aggregated susceptible trees, mass attacks grow in size and become capable of overwhelming formidable tree defenses. Even large healthy trees may be susceptible to colonization and mortality when beetle population density is high (Raffa et al. 2015, Bentz et al. 2010, Boone et al. 2011). Thus, water stress and beetle population density interact to influence whether individual trees are susceptible to bark beetles. When extreme or prolonged drought increases host tree vulnerability, bark beetle population growth rates increase, then become self-amplifying as greater beetle densities make additional host trees prone to successful mass attack (Stephenson et al. 2019, Bentz et al. 2010, Raffa et al. 2008, Boone et al. 2011).

R2.5: b. At times this ms conflates stand-level (forest structure) and tree-level (defense) processes. An example is L104: “Stand-scale measures of forest structure and composition thus paint a fundamentally limited picture of the mechanisms by which these forest characteristics affect bark beetle disturbance, but finer-grain information explicitly recognizing tree species, size, and local density should better capture the ecological processes underlying insect-induced tree mortality (Geiszler & Gara 1978, Mitchell & Preisler 1991, Preisler 1993, Kaiser et al. 2013).” The references I suggested that are now cited pertain to stand-level processes, which is good. But the omitted reference (Raffa & Berryman 1982) provides tree-level (i.e., finer-grain) mechanisms to help fill that ‘limited picture’. That paper’s finding that suppressed trees had weaker biochemical defenses than dominant/codominant trees, and that within suppressed trees crowding reduces defense, seems to be provide ‘finer-grain information’ consistent with the current ms’ findings about relationships between tree size and beetle-caused mortality. Also, an important paper that has helped clarify stand-level processes is Faccoli & Bernardinelli 2014.

We have tried to make a stronger distinction between tree- and stand-scale processes. One way we’ve done this is by more appropriately referring to “coarse-scale measures of forest structure/composition” as being fundamentally limiting for elucidating WPB dynamics rather than “stand-scale measures”. For instance, the fraction of a stand that is the host species is a “stand-scale measure” but still requires some finer-scale information (e.g., how many individual trees of each species) to characterize. Our intention is to suggest that “coarse-scale” measures (e.g., NDVI in a 3.5 km pixel as a proxy for total biomass in a stand) can only go so far in progressing our understanding of how WPB responds to forest structure/composition.

We also now cite Raffa & Barryman 1982 in reference to an individual-level tree response, and we now cite Faccoli & Bernardinelli 2014 when discussing insect damage in pure vs. mixed species forests (i.e., a stand-level process).

The particularly relevant paragraph now reads:

The interaction between forest structure and composition at both stand- and tree- scales also drives WPB activity. For instance, dense forest stands with high host availability may experience greater beetle-induced tree mortality because dispersal distances between potential host trees are shorter, which reduces predation of adults searching for hosts and facilitates higher rates of colonization (Miller and Keen 1960, Berryman 1982, Fettig et al. 2007). High host availability can also reduce the chance of individual beetles wasting their limited resources flying to and landing on a non-host tree (Moeck et al. 1981, Evenden et al. 2014). At a finer scale, a host tree's defensive capacity can depend on its canopy position, with reduced biochemical defenses in suppressed, crowded trees (Raffa and Berryman 1982). Coarse-scale measures of forest structure and composition can therefore only partially explain mechanisms affecting bark beetle disturbance. Finer-grain information is also needed that explicitly recognizes tree species, size, and local density, which better capture the ecological processes underlying insect-induced tree mortality (Geiszler and Gara 1978, Mitchell and Preisler 1991, Preisler 1993, Kaiser et al. 2013).

R2.6: *c. The discussion places strong emphasis on the negative main effect of host tree size, and how the relationship of host tree size to tree mortality varies with environmental stress. This is indeed new and important information for western pine beetle. However, restricting this discussion to western pine beetle makes the ramifications seem needlessly narrow for Nature Communications and detached from related systems. It misses an opportunity to update long-held views (conventional wisdom) of host size more generally by not pointing to the consistency of the current results with results on mountain pine beetle when analyses of host size effects are linked to tree defense physiology (Boone et al. 2011).*

R2.7: 3. Line 495: *It is probably better to say 'conventional wisdom' than 'wisdom'.*

We agree. We now say "conventional wisdom", though note that this sentence now reflects our updated analysis:

The positive main effect of host tree mean size on ponderosa mortality rates tracks the conventional wisdom on the dynamics of WPB in the Sierra Nevada, as well as other primary bark beetles (Fettig et al. 2016).

R2.8: 4. Line 171-172: Yes, western pine beetle outcompetes mountain pine beetle for live ponderosa pines, but because this paragraph describes population dynamics rather than tree mortality my comment about other competitors was meant to include beetle development success following tree death. Secondary bark beetles such as *Ips* and *D. valens*, wood borers, and antagonistic fungi can reduce primary beetle reproduction through competition, even despite some spatial and temporal separation.

We better understand the importance of this point and would also add that predation during dispersal will be additionally impactful to populations in a similar vein (i.e., describing population dynamics aside from tree mortality). We now say:

WPB population growth rates can, however, be reduced by competition with other beetle species cohabitating in the same host tree, and by predation during dispersal to seek a host [Miller 1960].

R2.9: 5. Line 565: You should clarify that lodgepole pine monocultures are from natural regeneration, as some readers might envision uniform plantings as with European spruce and loblolly pine. Also, it might protect you from criticism to say 'relative' monocultures or apply some other modifier.

Good point. We now say:

This challenge of classifying standing dead trees to species implies that a conifer forest systems with less bark beetle and tree host diversity, such as mountain pine beetle outbreaks in relative monocultures of naturally-occurring lodgepole pine forests in the Intermountain West, should be particularly amenable to the methods presented here even with minimal further refinement because dead trees will almost certainly belong to a single species and have succumbed to colonization by a single bark beetle species.

R2.10: 6. Line 472: Two more appropriate references than Raffa & Berryman 1987 about thinning are Fettig 2007 and Fettig et al. 2014, so replace the former with one or both of the latter two.

Done!

R2.11: *Nice work.*

Thanks!

Reviewer #3 (Remarks to the Author):

R3.1: *I reviewed the original submission of this manuscript, and appreciate the careful attention to all of the reviewer comments, including mine, in the revision. The methods are more clearly presented, and the method and results are very helpful and the manuscript will be an important contribution. My only suggestion is to reiterate my plea for presenting the mortality results as percent of 'plot' basal area in addition to the percent of numbers of trees:*

From my original review: "2. Proportion of tree mortality relative to total trees in a plot is also a very poor surrogate for the effects of that mortality on the ecosystem. It is not the number of trees killed or the relative number of trees that are killed that is important to ecosystem effects such as water use, streamflow, or carbon storage, it is the proportion of basal area that is lost. The use of proportional number of trees killed instead of the proportion of basal area killed may also have contributed to the odd results (tree density and mean height having a negative effect on mortality fraction), and compared to earlier studies. Again, basal area would be much more biologically meaningful than tree numbers."

I get that the aim was focused on individual trees, but trees come in widely different sizes, such that the effects of the number or percent of trees killed in a stand will vary widely with the size of those trees. Basal area takes care of this issue and would be more useful to forest managers and forest ecologists. Since the authors included basal area as an estimate of competition, I believe that the authors have the data to estimate this metric and provide more useful information than numbers of trees or percent of total trees. Managers would be less concerned if the mortality were focused in just small trees and very concerned if it focused in just large trees. Likely it is both, but perhaps with a different pattern than for tree numbers. Doing basal area mortality might be analogous to identifying age and race for human Covid-19 mortality--people are roughly the same size, but age and race give the public health folks for information for management.

Mike Ryan

Thank you for these further thoughts on this. We think we better understand your perspective now. Our tree height calibration (noted above) now shows that greater mean tree height increases the probability of host tree mortality (rather than decreases it), and we have also added an additional figure to help capture the “basal area consequences” of an increasing proportion of dead trees in our study. We tried a number of ways to capture basal area as a response variable (outlined below), but none produced satisfying modeling results (i.e., models that fit the data well), likely because of the challenge of measuring basal area from the air (using species-specific

allometric equations to convert height, which we did measure, to DBH, then to basal area). We also now include this caveat and limitation in our discussion.

If you'll indulge us, we'll also respond by first summarizing our understanding of your critique and then explaining our approach to addressing it. We do this a) to make sure we're on the same page and b) for the benefit of future readers of this peer review dialogue in case our work is recommended for publication.

Our understanding is that, if we were using individual tree level data with the probability of tree mortality as a response (i.e., a single trial binomial response, a.k.a. a Bernoulli distribution), then the coefficient estimates would tell us all we need to know (e.g., if tree height was associated with greater probability of tree mortality, then this tells us the bigger trees are dying, representing a greater loss of basal area). However, because we are using a plot-level response of probability of tree mortality (equating to the proportion of trees within a plot that we'd expect to die) then the coefficient estimates may or may not correspond to *which* of the trees within a plot are dying. E.g., if mean tree height was positively associated with a greater proportion of dead trees in a plot, it might be the case that the larger trees were the ones to die, but it might also be the case that larger trees within a plot led to the demise of their smaller neighbors rather than themselves.

We tried a number of approaches to model the proportion of dead basal area in a plot using the same set of predictors as we used for the number of dead trees conditional on the total number of host trees in a plot, but ultimately none of these efforts produced a model that fit well enough to generate inference with much confidence (i.e., the posterior predictive checks failed to mimic the observations whereas the posterior predictive checks for the model predicting the number of dead trees given the number of total hosts did an excellent job of generating data that looks like our observations).

- Zero-one-inflated beta distributed response of the proportion of total basal area that was dead, with both the zero/non-zero and one/non-one processes modeled as constants but the mean modeled using the same predictors as the paper's current main model.
 - (Up through 2000 warmup, 4000 samples, 4 chains, 0.95 adapt_delta; still 2 divergent transitions)
- Smithson and Virkuilen (2006) transformed proportion of total basal area that was dead modeled as a beta
- Smithson and Virkuilen (2006) transformed proportion of host basal area that was dead modeled as a beta
- Zero-one-inflated beta distributed response of the proportion of host basal area that was dead, with all three processes modeled using the same predictors as the paper's current main model

- Beta distributed response of proportion of host basal area that was dead, excluding pixels with 0 or 1
- Hurdle-gamma distributed response of the rate of dead basal area per total basal area
- Hurdle-lognormal distributed response of the amount of dead basal area with zero process modeled as a constant
- Hurdle-lognormal distributed response of the amount of dead basal area with zero process modeled using the same covariates as the # of dead trees given # of hosts
- Lognormal distributed response of the amount of dead host basal area, conditional on there being *some* dead basal area
- Zero-one-inflated beta distributed response of the proportion of host basal area that was dead, with all three processes modeled using the same predictors as the paper's current main model, but no Gaussian process of x , y by site
- Zero-one-inflated beta distributed response of the proportion of total basal area that was dead, with all three processes modeled using the same predictors as the paper's current main model, but no Gaussian process of x , y by site

Reviewer comments, third round –

Reviewer #2 (Remarks to the Author):

This revision addresses all of the concerns I raised in my prior reviews. They likewise appear to have addressed the other reviews as well. As before, the authors provided highly organized responses to my questions, which makes the paper's changes and the authors' thinking easy to follow. I think this paper will make a fine contribution to Nature Communications.

Reviewer #3 (Remarks to the Author):

I appreciate that the authors did the analysis I requested. Unfortunately, basal area did not work out statistically, likely because of the data available. Regardless, the study is an important contribution, and I recommend that it be accepted.
Mike Ryan